# Single-cell transcriptome analysis defines heterogeneity of the murine pancreatic ductal tree

Audrey M Hendley[1,2], Arjun A Rao[3,4†], Laura Leonhardt[1†], Sudipta Ashe[1†], Jennifer A Smith[1], Simone Giacometti[1], Xianlu L Peng[5,6], Honglin Jiang[7], David I Berrios[1], Mathias Pawlak[8], Lucia Y Li[1], Jonghyun Lee[1], Eric A Collisson[7], Mark S Anderson[1], Gabriela K Fragiadakis[3,4,9], Jen Jen Yeh[5,6,10], Chun Jimmie Ye[11], Grace E Kim[12], Valerie M Weaver[2], Matthias Hebrok[1*]

[1]Diabetes Center, University of California, San Francisco, San Francisco, United States; [2]Center for Bioengineering and Tissue Regeneration, University of California, San Francisco, San Francisco, United States; [3]CoLabs, University of California, San Francisco, San Francisco, United States; [4]Bakar ImmunoX Initiative, University of California, San Francisco, San Francisco, United States; [5]Department of Pharmacology, University of North Carolina at Chapel Hill, Chapel Hill, United States; [6]Lineberger Comprehensive Cancer Center, University of North Carolina at Chapel Hill, Chapel Hill, United States; [7]Division of Hematology and Oncology, Department of Medicine and Helen Diller Family Comprehensive Cancer Center, University of California, San Francisco, San Francisco, United States; [8]Evergrande Center for Immunologic Diseases, Harvard Medical School and Brigham and Women's Hospital, Boston, United States; [9]Department of Medicine, Division of Rheumatology, University of California, San Francisco, San Francisco, United States; [10]Department of Surgery, University of North Carolina at Chapel Hill, Chapel Hill, United States; [11]Parker Institute for Cancer Immunotherapy, San Francisco, United States; [12]Department of Pathology, University of California, San Francisco, San Francisco, United States

*For correspondence:
Matthias.Hebrok@ucsf.edu

†These authors contributed equally to this work

Competing interests: The authors declare that no competing interests exist.

**Abstract** To study disease development, an inventory of an organ's cell types and understanding of physiologic function is paramount. Here, we performed single-cell RNA-sequencing to examine heterogeneity of murine pancreatic duct cells, pancreatobiliary cells, and intrapancreatic bile duct cells. We describe an epithelial-mesenchymal transitory axis in our three pancreatic duct subpopulations and identify osteopontin as a regulator of this fate decision as well as human duct cell dedifferentiation. Our results further identify functional heterogeneity within pancreatic duct subpopulations by elucidating a role for geminin in accumulation of DNA damage in the setting of chronic pancreatitis. Our findings implicate diverse functional roles for subpopulations of pancreatic duct cells in maintenance of duct cell identity and disease progression and establish a comprehensive road map of murine pancreatic duct cell, pancreatobiliary cell, and intrapancreatic bile duct cell homeostasis.

## Introduction

Pancreatic duct cells, while a minority of the composition of the pancreas, play an integral role in secretion and transport of digestive fluid containing proenzymes synthesized by acinar cells, electrolytes, mucins, and bicarbonate. They can serve as a cell of origin for pancreatic ductal

adenocarcinoma (PDA) (*Bailey et al., 2016*; *Lee et al., 2019*) and have been implicated in the pathophysiology of multiple other diseases including cystic fibrosis (*Wilschanski and Novak, 2013*) and pancreatitis (*Apte et al., 1997*).

Heterogeneity of a cell type becomes increasingly important in the context of disease and regeneration since different subpopulations can be the driving forces behind pathogenesis. The function of exocrine pancreatic cells is required for survival, yet these cells exhibit limited regenerative capabilities in response to injury. Chronic pancreatitis (CP) is a risk factor for pancreatic cancer. The underlying mechanisms for PDA progression in CP patients are incompletely understood and are likely multifactorial, including both genetic and environmental insults (*Etemad and Whitcomb, 2001*). Studies have shown that cytokines and reactive oxygen species generated during chronic inflammation can cause DNA damage. It has been hypothesized that pancreatic cells might acquire DNA damage in the protooncogene *KRAS* or tumor suppressor genes *TP53* or *CDKN2A*, thereby accelerating malignant transformation (*Whitcomb and Greer, 2009*; *Dhar et al., 2015*). Thus, it is imperative to understand the mechanisms by which DNA damage occurs in the setting of CP. Duct obstruction is one cause of CP, and the ability of ductal cells to acquire DNA damage in the setting of CP is incompletely understood.

In this report, we conducted single-cell RNA-sequencing (scRNA-seq) on homeostatic murine pancreatic duct, intrapancreatic bile duct, and pancreatobiliary cells using a DBA$^+$ lectin sorting strategy, and present a high-resolution atlas of these murine duct cells. By extensively comparing our subpopulations to previously reported mouse and human pancreatic duct subpopulations (*Qadir et al., 2020*; *Baron et al., 2016*; *Grün et al., 2016*), we both corroborate several previous findings and identify and validate novel duct cell heterogeneity with unique functional properties including roles for subpopulation markers in CP. Our findings suggest that multiple duct subpopulations retain progenitor capacity, which is influenced by expression of markers driving subpopulation identity.

## Results

### scRNA-seq identifies multiple pancreas cell types with DBA lectin sorting

Previously reported subpopulations of murine pancreatic duct cells were identified by single-cell analysis of pancreatic cells obtained using an islet isolation procedure; thus, exocrine duct cells were of low abundance (*Baron et al., 2016*). To circumvent this issue, we employed a DBA lectin sorting strategy that has been extensively used to isolate and characterize all murine pancreatic duct cell types (*Beer et al., 2016*; *Reichert et al., 2013*), to investigate murine duct heterogeneity. We isolated live DBA$^+$ cells from the pancreata of four adult female C57BL/6J littermates and performed scRNA-seq on the pooled cells using the 10X Genomics platform (*Figure 1A* and *Figure 1—figure supplement 1A*). After filtering out doublets and low-quality cells (defined by low transcript counts), our dataset contained 6813 cells. Clustering analysis identified 16 distinct cell populations with an average of 5345 transcripts per cell and 1908 genes per cell (*Figure 1B* and *Figure 1—source data 1*). Significantly differentially expressed genes (DEGs) when comparing a cluster to all other clusters are listed in *Figure 1—source data 2*. Annotation of these 16 clusters was accomplished by analysis of known markers (*Figure 1B–D*). Our dataset comprises 2 populations of ductal cells, a cluster of endothelial cells, 1 cluster of fibroblasts, and 12 immune cell clusters. As expected, murine endocrine and acinar cells are not present in our dataset because they are not DBA$^+$ cells. Gene and transcript counts for each cluster are shown in *Figure 1—figure supplement 1B*. We identified DBA$^+$Collagen I$^+$ fibroblasts and DBA$^+$CD45$^+$ immune cells by immunofluorescence (IF). CD31$^+$ endothelial cells are not DBA$^+$. Their presence in our dataset might be explained by the close juxtaposition of pancreatic duct cells with endothelial cells throughout the murine pancreas (*Figure 1—figure supplement 1C*).

### Subpopulations of ductal cells are characterized by unique gene signatures and regulation of pathways

To get a better understanding of duct cell heterogeneity, we generated an Uniform Manifold Approximation and Projection (UMAP) plot using all duct cells (clusters 0 and 8), which revealed six

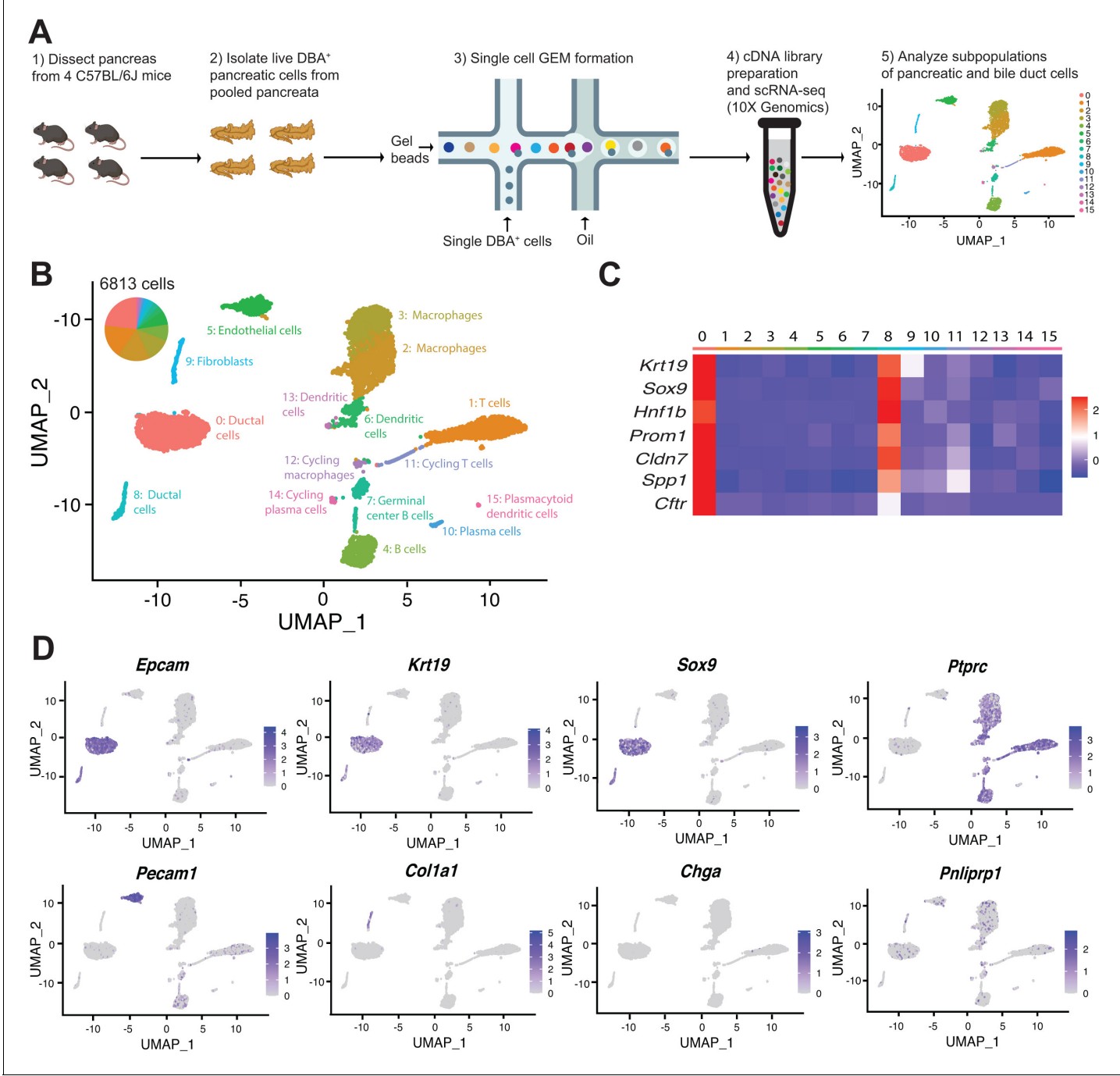

**Figure 1.** Transcriptomic map of DBA[+]pancreatic cells. (**A**) Schematic of experiment workflow. (**B**) The Uniform Manifold Approximation and Projection depicts murine pancreatic DBA[+] cells obtained using the protocol. (**C**) A matrix plot shows average expression of ductal cell markers in all clusters, identifying clusters 0 and 8 as ductal cells. (**D**) Feature plots illustrate markers of various cell types including epithelial (*Epcam*), ductal (*Krt19* and *Sox9*), CD45[+] immune cells (*Ptprc*), endothelial cells (*Pecam1*), fibroblasts (*Col1a1*), endocrine cells (*Chga*), and acinar cells (*Pnliprp1*). We observed low-level expression of acinar cell markers uniformly across all clusters that is likely contaminating acinar cell mRNA.

The online version of this article includes the following source data and figure supplement(s) for figure 1:

**Source data 1.** Number of cells and average number of genes and transcripts in all DBA[+] clusters.
**Source data 2.** Differentially expressed genes in all DBA[+] clusters.
**Figure supplement 1.** Features of DBA[+] (clusters 0–15) and ductal (clusters 0–5) cells.

distinct ductal clusters. Annotation of each duct cluster was accomplished using DEGs, ingenuity pathways analysis (IPA) and upstream regulator analysis, and marker assessment in murine and human pancreas (*Figure 2A–D*, *Figure 1—figure supplement 1D, E, Scheme 1*, and *Figure 2— source data 1–3*). Gene and transcript counts for each cluster are shown in *Figure 1—figure supplement 1F* and *Figure 2—source data 4*. We observed variable expression of known ductal markers within clusters. Notably, fewer murine duct cells express the transcription factor *Hnf1b* when compared to *Sox9*. This observation is in contrast to a previous report demonstrating a similar prevalence of adult murine HNF1B+ and SOX9+ duct cells, which might be explained by different ductal cell isolation methods (*Figure 1—figure supplement 1G*; *Rezanejad et al., 2018*).

Cluster 0 contains the most cells of all duct clusters in the dataset (*Figure 2—source data 4*). A gene that positively regulates Ras signaling *Mmd2*, the voltage-gated potassium channel protein encoded by *Kcne3*, as well as the ATP-binding cassette (ABC) transporter chloride channel protein encoded by *Cftr*, were significantly upregulated in cluster 0 when compared to all other ductal clusters (*Figure 2C* and *Figure 2—source data 1*). Notably, cluster 0 shows upregulation or activation of multiple genes whose alteration play important roles in the pathophysiology of human pancreatic diseases such as *CFTR* for hereditary CP (*Raphael and Willingham, 2016*) and *TGFB2* and *CTNNB1* for pancreatic cancer (*Shen et al., 2017*; *Gordon et al., 2008*; *Heiser et al., 2008*; *Figure 2— source data 1* and *3*).

To validate gene expression patterns and determine the location of cluster 0 cells within the hierarchical pancreatic ductal tree (*Reichert and Rustgi, 2011*), we next examined expression of select significantly DEGs. *Gmnn*, an inhibitor of DNA replication, was expressed in both clusters 0 and 2, so we decided to examine histologically and were surprised to find rare GMNN protein expression, which was in contrast to the widespread RNA expression depicted by the feature plot (*Figure 2— figure supplement 1A*). After examining more than 1500 main pancreatic duct cells from five donors, we were unable to find a GMNN-positive cell, indicating very low or absent expression of GMNN in human main pancreatic ducts. *Spp1*, which encodes osteopontin, and *Wfdc3*, which are significantly DEGs in both clusters 0 and 2, show cytoplasmic protein expression in all mouse and human pancreatic duct types (*Figure 2—figure supplement 1B, C* and *Supplementary file 1*).

Cells in cluster 1 have significantly upregulated expression of the exosome biogenesis gene *Rab27b* as well as *Ppp1r1b* that encodes for a molecule with kinase and phosphatase inhibition activity (*Figure 2A–C* and *Figure 2—source data 1*). IPA results suggested an enrichment in molecules regulating Calcium Transport I (*Figure 2D* and *Figure 2—source data 2*). IPA upstream regulator analysis predicted an activated state for the transcriptional regulator *Smarca4* and the two growth factors TGFB1 and GDF2 (*Figure 2—source data 3*). Intracellular calcium signaling in pancreatic duct cells is an important regulator of homeostatic bicarbonate secretion (*Maléth and Hegyi, 2014*). *PPP1R1B*, *SMARCA4*, and *TGFB1* have well-described roles in the pathogenesis of pancreatic cancer (*Roy et al., 2015*; *David et al., 2016*; *Tiwari et al., 2020*). We observed expression of markers of cluster 1, *Anxa3* and *Pah*, which are also DEGs in cluster 4, to have cytoplasmic protein expression in all mouse and human pancreatic duct types (*Figure 2—figure supplement 2A, B* and *Supplementary file 1*). Co-staining of CFTR, a marker of cluster 0, and ANXA3 shows both overlapping and non-overlapping patterns of expression in human intercalated ducts, validating the heterogeneity observed in our murine pancreatic duct dataset in human pancreatic duct cells (*Figure 2— figure supplement 2C*).

Cluster 2 is characterized by low level or lack of expression of multiple ductal cell markers (*Cftr, Kcne3, Sparc, Mmd2, Krt7*) found in other clusters (*Figure 2B, C* and *Figure 1—figure supplement 1G*). Cluster 2 has the lowest average expression of total genes and transcripts (*Figure 1—figure supplement 1F* and *Figure 2—source data 4*). We therefore posit that cluster 2 represents a stable, fairly transcriptionally and metabolically inactive duct cell subpopulation when compared to other duct clusters. Cluster 3 cells are located almost entirely within cluster 8 of the UMAP containing 16 DBA+ clusters (*Figure 1—figure supplement 1E*). This, along with high expression of genes regulating cilia biogenesis (*Foxj1, Cfap44, Tuba1a*), led to the identification of cluster 3 as intrapancreatic bile duct cells (*Figure 2A–C* and *Figure 2—source data 1*). Expression of cilia biogenesis genes is more prominent in intrapancreatic bile duct cells when compared to pancreatic duct cells (*Figure 2— figure supplement 2D*, *Figure 2—source data 1*, and data not shown).

Cells in cluster 4 have significantly higher expression of *Tgfb3* and *Dclk1* when compared to all other ductal clusters (*Figure 2C* and *Figure 2—source data 1*). DCLK1 labels tuft cells that are

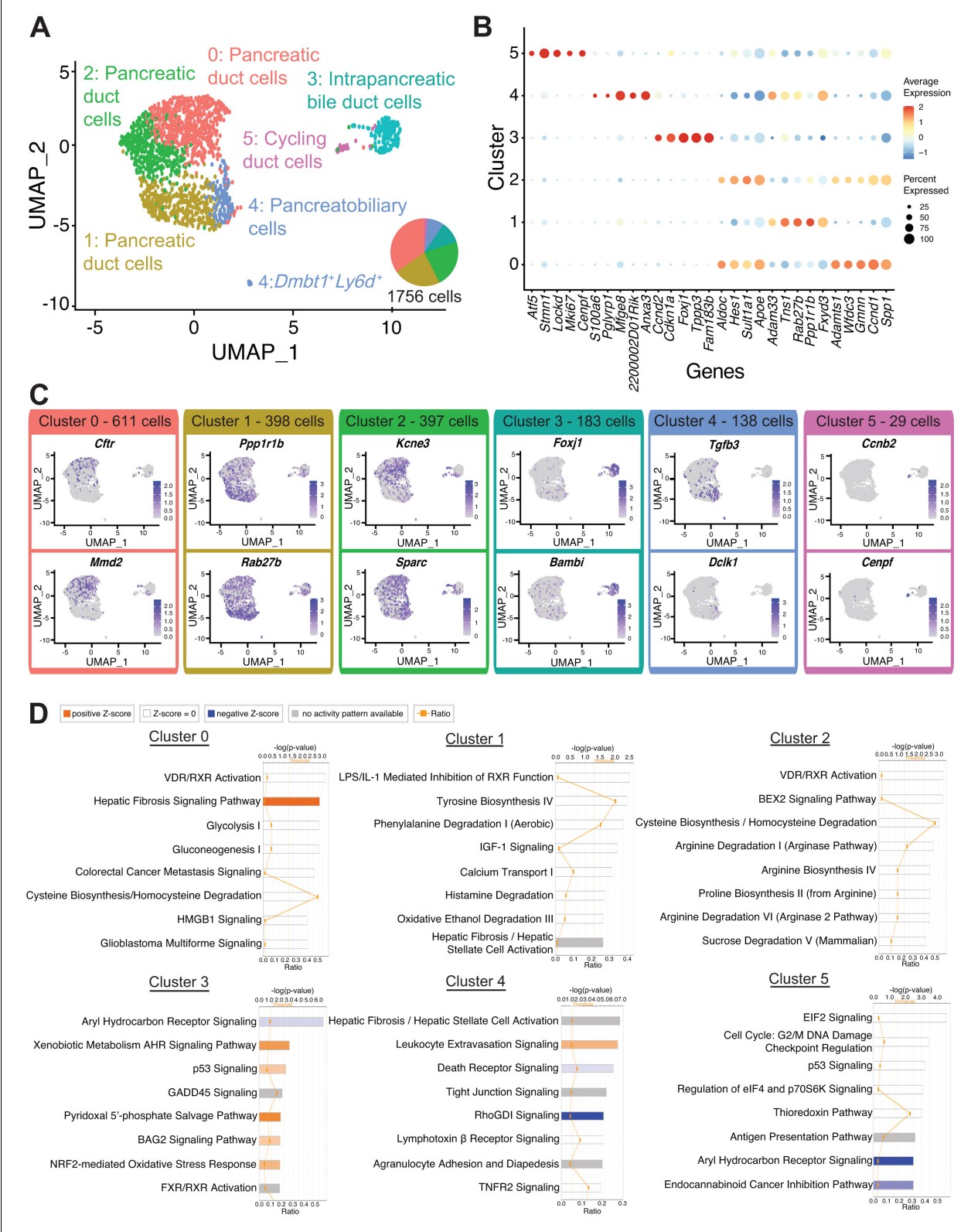

**Figure 2.** Transcriptomic map of DBA⁺pancreatic duct cells. (**A**) Uniform Manifold Approximation and Projection (UMAP) depicts identity of clusters. (**B**) The dot plot shows the top five significantly differentially expressed genes (DEGs) with the highest fold change for each cluster. (**C**) Feature plots show expression of significantly DEGs for clusters 0, 1, 3, 4, and 5. Cluster 2 is characterized by lack of or low-level expression of significantly DEGs found in other clusters. (**D**) Ingenuity pathways analysis (IPA) results show the top eight deregulated pathways when comparing a cluster to all other clusters. The

*Figure 2 continued on next page*

*Figure 2 continued*

ratio line indicates the fraction of molecules significantly altered out of all molecules that map to the canonical pathway from within the IPA database. A positive z-score represents upregulation, and a negative z-score indicates downregulation of a pathway in that cluster when compared to all other clusters. A gray bar depicts significant overrepresentation of a pathway, the direction of which cannot yet be determined.

The online version of this article includes the following source data and figure supplement(s) for figure 2:

**Source data 1.** Differentially expressed genes in all DBA+ duct clusters.
**Source data 2.** Ingenuity pathways analysis results for all DBA+ duct clusters.
**Source data 3.** Ingenuity pathways analysis upstream regulator analysis results for all DBA+ duct clusters.
**Source data 4.** Number of cells and average number of genes and transcripts in all DBA+ duct clusters.
**Figure supplement 1.** IHC illustrates expression of markers in clusters 0 and 2 in the mouse and human ductal tree.
**Figure supplement 2.** IHC and IF depict expression of markers in clusters 1, 3, 4, and 5 in mouse and human pancreas duct cells.
**Figure supplement 3.** Characteristics of intrapancreatic bile duct and pancreatobiliary cells.
**Figure supplement 3—source data 1.** Differentially expressed genes comparing duct cluster 3 vs. 4.
**Figure supplement 3—source data 2.** Ingenuity pathways analysis results comparing duct cluster 3 vs. 4.
**Figure supplement 3—source data 3.** Ingenuity pathways analysis upstream regulator analysis comparing duct cluster 3 vs. 4.

present in normal murine intrapancreatic bile ducts, pancreatobiliary ductal epithelium (*DelGiorno et al., 2014*), and rare normal murine pancreatic duct cells (*Westphalen et al., 2016*). YAP1, a transcriptional regulator essential for homeostasis of biliary duct cells (*Pepe-Mooney et al., 2019*), was predicted to be in an activated state by IPA upstream regulator analysis (*Figure 2—*

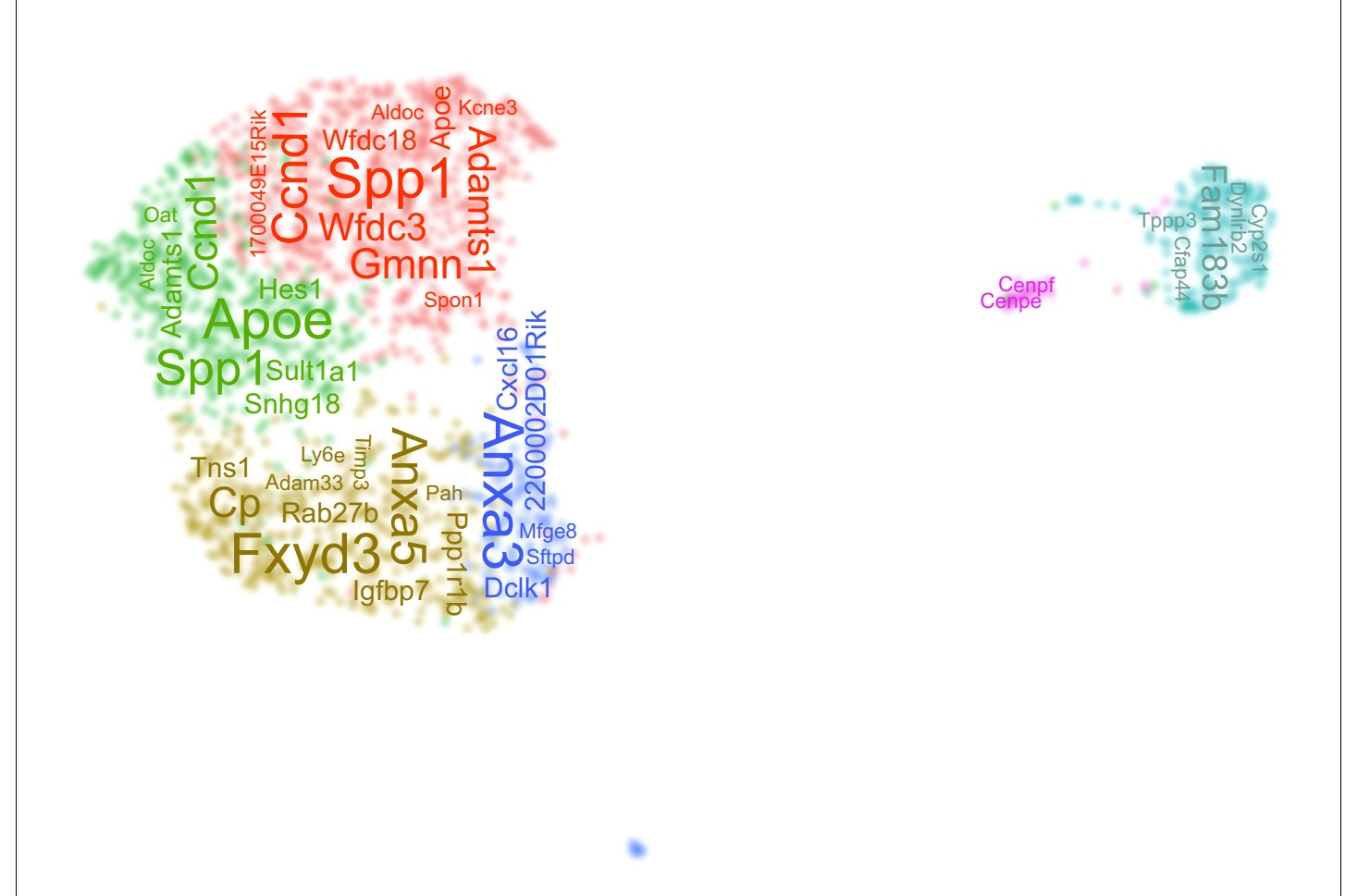

**Scheme 1.** A word cloud depicts the top differentially expressed genes in each duct cluster.

*source data 3*). Cluster 4 also contained a small population (13 cells) of *Dmbt1* and *Ly6d*-expressing cells previously identified in extrahepatic biliary epithelium (*Pepe-Mooney et al., 2019*; *Figure 2—figure supplement 3A*). These 13 cells appeared as a small population separate from other cells in cluster 4 in the UMAP (*Figure 2A*). Similar to the IF validation reported for extrahepatic biliary epithelial cells (BECs) (*Pepe-Mooney et al., 2019*), our IF assessment of CXCL5, another marker of the *Dmbt1* and *Ly6d*-expressing subpopulation, showed a greater abundance of these cells than what would be expected given the number identified in the clustering analysis (*Rezanejad et al., 2018*). It is possible that this cell type is sensitive to single-cell dissociation. Cells in cluster 4 are juxtaposed to pancreatic duct cells (clusters 0, 1, and 2) in the UMAP, suggesting transcriptional commonalities with pancreatic duct cells. In addition, *Dmbt1* and *Ly6d*-expressing cells are present in cluster 4, suggesting a bile duct identity. Based on these shared features of bile and pancreas ducts, we postulate that cluster 4 contains pancreatobiliary duct cells.

Replicating duct cells are characterized by high expression of *Mki67*, *Cenpf*, and *Cenpe* and comprise 1.65% of all duct cells in our dataset (*Figure 2A–C*, *Figure 2—figure supplement 2D*, and *Figure 2—source data 1*). Consistent with previous reports (*Moin et al., 2017*; *Butler et al., 2010*), pancreatic duct cells are fairly mitotically inactive.

Summarily, our high-resolution single-cell analysis has identified the substructure of murine pancreatic duct cells and characterized pancreatobiliary and intrapancreatic bile duct cells.

## Comparison of clusters defines heterogeneity within duct subpopulations

We next sought to determine the relationships between duct clusters by examining their similarities and differences. Dendrogram analysis, Pearson's correlation, and DEGs revealed close relationships between clusters 0 and 2 as well as clusters 1 and 4 (*Figure 3A, B* and *Figure 3—source data 1*). Comparison of clusters 0 and 2 showed only nine significant DEGs, suggesting a shared core gene expression program (*Figure 3C, D*). Overrepresentation of molecules regulating the cell cycle was observed in cluster 0 when compared to cluster 2 (*Figure 3E*). The DEGs upregulated in cluster 0 promote duct cell function (*Cftr*, *Tuba1a*, *Kcne3*), suggesting that cluster 0 comprises workhorse pancreatic duct cells (*Hayashi and Novak, 2013*).

When comparing pancreatobiliary cells of cluster 4 to pancreatic duct cells in cluster 1, one of the most striking differences is the enrichment in expression of genes regulating assembly of cell junctions including tight junctions, epithelial adherens junction signaling, regulation of actin-based motility by Rho, and actin cytoskeleton signaling. A strong network of stress fibers, comprising actin filaments, myosin II, and other proteins, that function in bearing tension, supporting cellular structure, and force generation may be important for pancreatobiliary cell function and maintenance (*Figure 3F–H* and *Figure 3—source data 2* and *3*; *Burridge and Wittchen, 2013*; *Tojkander et al., 2012*). Cluster 4: $Dmbt1^+Ly6d^+$ cells are characterized by strong upregulation of pathways regulating xenobiotic metabolism when compared to all other cluster 4 cells, suggesting a prominent role for these cells in the bile acid and xenobiotic system (BAXS) (*Figure 3I–K* and *Figure 3—source data 2* and *3*; *Dubitzky et al., 2013*). Comparison of intrapancreatic bile duct cells and pancreatobiliary cells showed many unique features of these populations including upregulation of EIF2 signaling in pancreatobiliary cells and upregulation of coronavirus pathogenesis pathway in intrapancreatic bile duct cells (*Figure 2—figure supplement 3B–D* and *Figure 2—figure supplement 3—source data 1–3*).

## Pancreatobiliary cells express a gene signature enriched in several targets of the Hippo signaling pathway YAP

Two subpopulations of adult murine hepatic homeostatic BECs, A and B, have been previously described (*Pepe-Mooney et al., 2019*). To determine if these subpopulations are present in intrapancreatic bile duct (cluster 3) and pancreatobiliary cells (cluster 4), we aligned our dataset with an adult hepatic murine BEC scRNA-seq dataset comprising 2344 homeostatic BECs (*Pepe-Mooney et al., 2019*). Intrapancreatic bile duct and pancreatobiliary cells aligned well with hepatic BECs, with no apparent batch effect (*Figure 3—figure supplement 1A*). Intrapancreatic bile duct cells primarily cluster together with hepatic BECs expressing subpopulation B genes, and pancreatobiliary cells primarily cluster together with hepatic BECs expressing subpopulation A genes

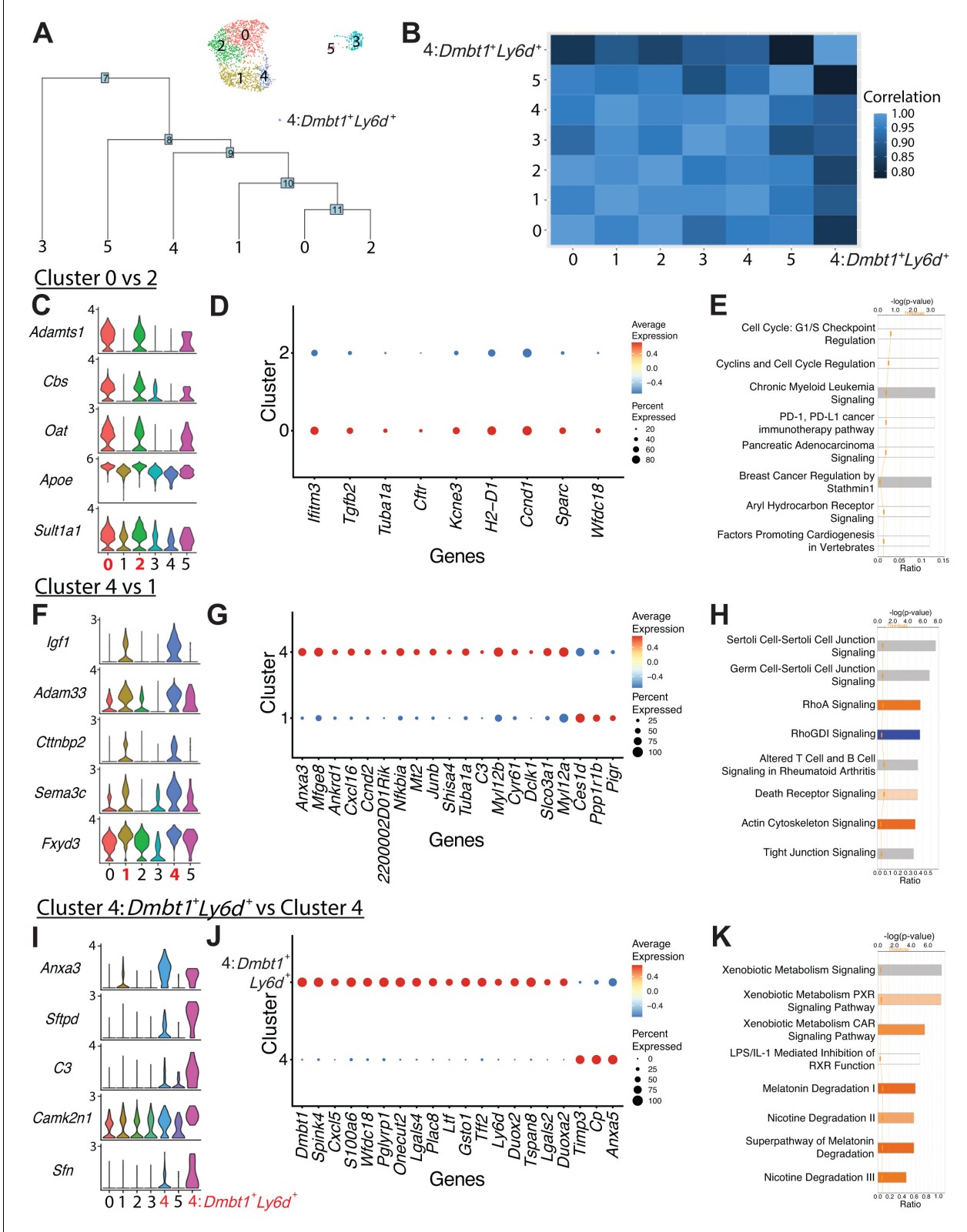

**Figure 3.** Comparison of ductal clusters 0 vs. 2, 4 vs. 1, and 4 vs. 4:*Dmbt1*+*Ly6d*+. (**A**) The cluster dendrogram created using dims (used to define the cluster) shows the Euclidean relationships between clusters. The tree is calculated in the principal component analysis space. The genes used to define the tree were set as the variable features of the object. (**B**) Pearson's correlation calculated using average gene expression is depicted. (**C**) Stacked violin plots show five differentially expressed genes (DEGs) sharing similar expression patterns in clusters 0 and 2. (**D**) The dot plot shows all nine DEGs

*Figure 3 continued on next page*

*Figure 3 continued*

found when comparing cluster 0 vs. 2. (E) The top eight altered pathways from ingenuity pathways analysis (IPA) comparing cluster 0 vs. 2 are depicted. (F) Stacked violin plots show five DEGs sharing similar expression patterns in clusters 4 and 1. (G) The dot plot shows the top 20 DEGs ranked by fold change when comparing cluster 4 vs. 1. (H) The top eight deregulated pathways from IPA comparing cluster 4 vs. 1 are depicted. (I) Stacked violin plots of five DEGs sharing similar expression patterns in clusters 4:*Dmbt1*+*Ly6d*+ and 4. (J) The dot plot shows the top 20 DEGs ranked by fold change when comparing clusters 4:*Dmbt1*+*Ly6d*+ and 4. (K) The top eight changed pathways from IPA comparing clusters 4:*Dmbt1*+*Ly6d*+ and 4 are depicted.

The online version of this article includes the following source data and figure supplement(s) for figure 3:

**Source data 1.** Differentially expressed genes comparing duct cluster 0 vs. 2, duct cluster 4 vs. 1, and duct cluster 4-Dbmt1+Ly6d+ vs. 4.

**Source data 2.** Ingenuity pathways analysis results comparing duct cluster 0 vs. 2, duct cluster 4 vs. 1, and duct cluster 4-Dbmt1+Ly6d+ vs. 4.

**Source data 3.** Ingenuity pathways analysis upstream regulator analysis comparing duct cluster 0 vs. 2, duct cluster 4 vs. 1, and duct cluster 4-Dbmt1+-Ly6d+ vs. 4.

**Figure supplement 1.** Alignment to an adult murine hepatic biliary epithelial cell (BEC) dataset.

**Figure supplement 1—source data 1.** Number of cells and average number of genes and transcripts for merged BEC–DBA+ duct clusters 3 and 4 dataset. BEC: biliary epithelial cell.

**Figure supplement 1—source data 2.** Differentially expressed genes in merged BEC–DBA+ duct clusters 3 and 4 dataset. BEC: biliary epithelial cell.

**Figure supplement 2.** Analysis of pancreas duct cells during development.

**Figure supplement 3.** Comparison of DBA+ lectin sorted mouse pancreas duct subpopulations to ALK3+ human pancreas duct subpopulations.

(*Figure 3—figure supplement 1B–G* and *Figure 3—figure supplement 1—source data 1* and *2*). The subpopulation A expression signature contains many genes significantly enriched as YAP targets, a signature that has been previously proposed to reflect a dynamic BEC state as opposed to defining a unique cell type (*Pepe-Mooney et al., 2019*).

## Ductal subpopulations are conserved and evident during pancreas development

To investigate whether pancreas ductal subpopulations become evident during development, we analyzed 10X Genomics single-cell published datasets of epithelial-enriched pancreas cells at E12.5, E14.5, and E17.5 (*Byrnes et al., 2018*). We found distinct subpopulations of ductal cells that notably overlap in expression of key marker genes associated with adult pancreas ductal subpopulations (*Figure 3—figure supplement 2A–L*). As we expected, clear patterns of marker gene expression associated with adult clusters manifest at later stages of development (*Figure 3—figure supplement 2D, H, L*). Since the developmental biology samples were obtained from Swiss Webster mice, our results suggest the subpopulations of adult pancreas duct cells we describe in C57BL/6J mice are conserved.

## DBA+ lectin murine pancreas sorting identifies previously missed ductal subpopulations

To determine the novelty of adult duct cell heterogeneity manifested using DBA+ lectin sorting of murine pancreas, we next compared our DBA+ murine pancreatic ductal clusters to previously reported subpopulations of mouse and human pancreas duct cells. Using inDrop and an islet isolation pancreas preparation, *Baron et al., 2016* identified the substructure of mouse and human pancreatic duct cells (*Baron et al., 2016*). Two subpopulations of mouse pancreatic duct cells characterized by expression of *Muc1* and *Tff2* (subpopulation 1) and *Cftr* and *Plat* (subpopulation 2) were described. While *Cftr* expression is characteristic of our cluster 0 (*Figure 2C*), *Muc1*, *Tff2*, and *Plat* expression did not typify any murine DBA+ pancreatic duct subpopulation (*Figure 3—figure supplement 2M*). Two subpopulations were similarly described for human pancreas duct cells characterized by expression of (1) *TFF1*, *TFF2*, *MUC1*, *MUC20*, and *PLAT* and (2) *CFTR* and *CD44*. *Tff1* is not expressed in murine DBA+ ductal cells (clusters 0–5). *Cd44* is significantly upregulated in pancreatobiliary cells, and *Muc20* as well as *Tff2* are significantly upregulated in 4:*Dmbt1*+*Lyd6*+ cells (*Figure 2—source data 1*, *Figure 3—source data 1*, and *Figure 3—figure supplement 2M, N*). Dominic *Grün et al., 2016* previously reported four subpopulations of human pancreatic duct cells characterized by expression of *CEACAM6*, *FTH1*, *KRT19*, and *SPP1* using an islet isolation pancreas preparation and the CEL-seq protocol (*Grün et al., 2016*). While *Spp1* is significantly upregulated in DBA+ pancreas duct clusters 0 and 2, *Fth1* does not characterize any murine DBA+ pancreas duct population, and *Krt19* is significantly upregulated in pancreatobiliary cells (*Figure 2—source data 1*,

*Figure 1—figure supplement 1G*, and *Figure 3—figure supplement 2O*). *CEACAM6* has no mouse homolog. The differences in pancreatic ductal subpopulation identification may be due to single-cell methodology (inDrop, CEL-seq, and 10X Genomics), pancreas preparation method (islet isolation vs. DBA$^+$ lectin sorting), differences in ductal cell numbers analyzed, or potential differences between mouse and human duct cells.

Six subpopulations of human pancreatic duct cells have been described using the 10X Genomics platform based on sorting for *BMPR1A/ALK3* (*Qadir et al., 2020*). Using AddModuleScore in Seurat, we calculated a score comparing each of our murine duct clusters to the human ALK3$^+$ clusters (*Figure 3—figure supplement 3A–F*; *Alshetaiwi et al., 2020*). Murine pancreatic duct clusters 0–2 had the highest scores when compared to human ALK3$^+$ clusters 1 (OPN$^+$stress/harboring progenitor-like cells) and 2 (TFF1$^+$ activated/migrating progenitor cells). Murine pancreatobiliary cells (cluster 4) scored the highest when compared to the human ALK3$^+$ cluster 3 (AKAP12$^+$ small ducts). The human ALK3$^+$ cluster 4 (WSB1$^+$ centroacinar cells) did not distinguishably overlap with any DBA$^+$ mouse pancreas ductal clusters. DBA is expressed in murine centroacinar/terminal ducts as early as 3 weeks of age (*Stanger et al., 2005*), thus these cells would be expected to be present in our dataset (*Beer et al., 2016*). Examination of expression of centroacinar/terminal ductal cell markers *Hes1* (*Kopinke et al., 2011*), *Aldh1a1* (*Rovira et al., 2010*), and *Aldh1b1* (*Mameishvili et al., 2019*) in our dataset showed broad expression enriched in either clusters 0 and 2 (*Hes1* and *Aldh1b1*) or clusters 1 and 4 (*Aldh1a1*), rather than a distinct subpopulation as is seen in the ALK3$^+$ human scRNA-seq pancreas duct dataset. *Aldh1a7* is negligibly expressed in murine duct clusters 0–5 (*Figure 3—figure supplement 3G*). Unlike in mouse DBA$^+$ pancreas duct clusters, the human ALK3$^+$ dataset contains two ducto-acinar subpopulations characterized by expression of genes enriched in acinar cells. To assess the presence of ducto-acinar cells in adult murine pancreas, we performed immunolabeling for markers of the ALK3$^+$ human ducto-acinar clusters 5 (*CPA1*) and 6 (*AMY2A* and *AMY2B*). Although ducto-acinar cells, like centroacinar/terminal ductal cells, do not define a unique cluster in our DBA$^+$ murine duct subpopulations, we identified DBA$^+$CPA1$^+$ and DBA$^+$α-amylase$^+$ ducto-acinar cells in adult murine pancreas (*Figure 3—figure supplement 3H*). Taken together, these data suggest that murine centroacinar/terminal ductal and ducto-acinar cells are largely transcriptionally homogenous with other murine duct cell types.

## RaceID3/StemID2 suggest murine DBA$^+$ duct cluster 0 and 2 cells are the most progenitor-like

Given the close relationships observed between DBA$^+$ duct clusters 0 and 2 as well as 1 and 4, we next assessed differentiation potential using RaceID3/StemID2 to predict cell types, lineage trajectories, and stemness (*Herman et al., 2018*). Unsupervised clustering with RaceID3 generated 17 clusters. RaceID3 clusters with 10 cells or less were removed from subsequent analyses, and Seurat duct clusters 3 and 5 are not included in this analysis (*Figure 4A, B*). RaceID3 clusters with the highest StemID2 score correlate to cells present in Seurat duct clusters 0 and 2 (*Figure 4C* and *Figure 5—figure supplement 1A, B*). The variable StemID2 scores observed for cells within Seurat duct clusters 0, 1, 2, and 4 suggest distinct stages of differentiation or maturation. Consistent with previous literature, the pancreatic ductal cell progenitor niche is not restricted to a single cluster (*Qadir et al., 2020*).

## Pseudotime ordering identifies an epithelial-mesenchymal transition (EMT) axis in pancreatic duct cells

To further examine the lineage relationships among pancreas duct subpopulations, we ordered cells in pseudotime based on their transcriptional similarity (*Cao et al., 2019*). Monocle 3 analysis suggested DBA$^+$ duct clusters 3 and 5 were disconnected from the main pseudotime trajectory, so we focused our analysis on DBA$^+$ duct clusters 0, 1, 2, and 4 (*Figure 5—figure supplement 1C*). Because RaceID3/StemID2 analysis suggested Seurat clusters 0 and 2 have the highest StemID scores, we started the pseudotime ordering beginning with cluster 0 as Seurat clusters 0 and 2 are juxtaposed in the Monocle 3 clustering (*Figure 4D, E* and *Figure 5—figure supplement 1D*).

In Monocle 3 analysis, genes with similar patterns of expression that vary over time across the pseudotime trajectory are coalesced into modules (*Figure 5A*). We performed IPA and upstream regulator analysis, a pairwise comparison, comparing select clusters within a module to analyze the

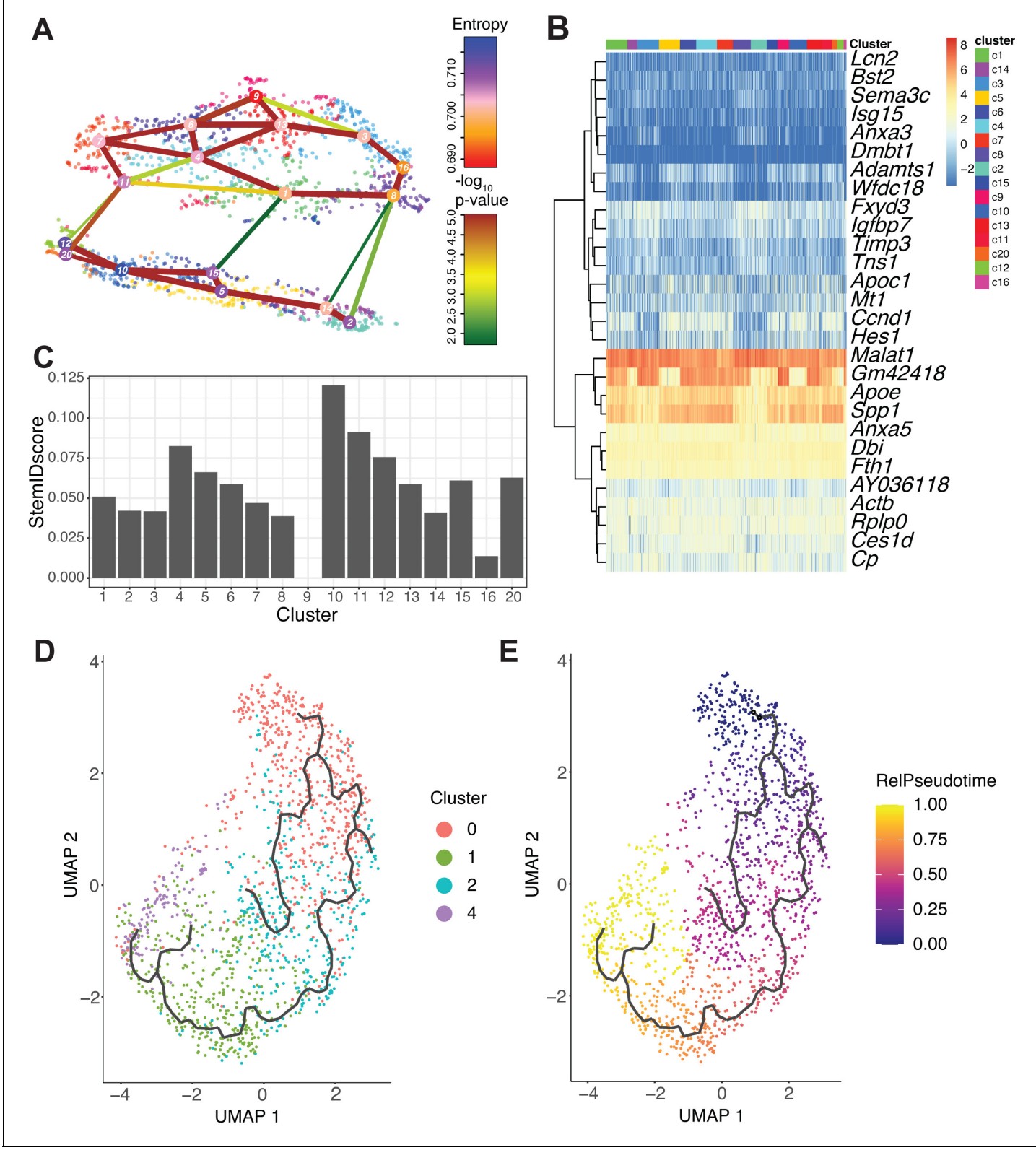

**Figure 4.** RaceID3/StemID2 predict clusters 0 and 2 have the highest progenitor potential. (**A**) The lineage tree inferred by StemID2 is shown in the RaceID3 clusters. Node color represents the level of transcriptome entropy, edge color describes the level of significance, and edge width describes link score. (**B**) Heat map depicts expression of top five differentially expressed genes in RaceID3 clusters with false discovery rate < 0.01 and fold change > 1.2. (**C**) StemID2 scores for RaceID3 clusters are graphed. (**D**) Monocle 3 clustering of murine DBA[+] duct clusters 0, 1, 2, and 4 is depicted. (**E**)

*Figure 4 continued on next page*

*Figure 4 continued*

Each cell's relative pseudotime value is depicted that is a measurement of the distance between its position along the trajectory and the starting point (cluster 0).

gene expression changes along the pseudotime trajectory (*Figure 5B–D* and *Figure 5—source data 1–3*). Examination of pathways deregulated in modules 4 and 14 showed a shift in the molecules driving the xenobiotic metabolism CAR signaling pathway. The xenobiotic nuclear receptor CAR is an important sensor of physiologic toxins and plays a role in their removal (*Timsit and Negishi, 2007*). The genes highlighted in the xenobiotic metabolism CAR signaling pathway were *Aldh1b1*, *Aldh1l1*, *Gstt2/Gstt2b*, *Hs6st2*, and *Ugt2b7* for clusters 0 and 2 and *Aldh1a1*, *Fmo3*, *Gstm1*, and *Sod3* for cluster 1, suggesting that these clusters might respond differently when exposed to toxins or play heterogenous roles in endogenous toxin elimination (*Figure 5B, C*).

Regulation of the epithelial-mesenchymal transition by growth factors pathway was upregulated in cluster 1 when compared to cluster 0 in module 34. Molecules altered in this pathway play variable roles in promoting the epithelial or mesenchymal state and include *Fgf12*, *Fgfr2*, *Fgfr3*, *Pdgfc*, and *Smad3* (*Figure 5D*). When comparing clusters 0 and 1, examination of EMT markers *Vim* and *Cdh1* showed a stronger probability of expression of *Cdh1* in cluster 1 and a stronger probability of expression of *Vim* in cluster 0 (*Figure 5E*). Using IF, we detected vimentin[+], SNAI1[+], and fibronectin[+] ductal cells in both mouse and human pancreas, providing evidence for this EMT axis (*Figure 5F*, *Figure 5—figure supplement 1E*, and data not shown).

## Osteopontin is required for mature human pancreas duct cell identity

Our analysis thus far reveals multiple transcriptional programs expressed by murine pancreatic duct cells and predicts possible lineage relationships among them. Amidst the duct subpopulation markers, *Spp1* and *Anxa3* caught our eye due to their known roles in pancreatic cancer progression (*Kolb et al., 2005*; *Adams et al., 2019*; *Wan et al., 2020*); however, their functions in normal pancreatic duct epithelium have not been fully explored. *Spp1*, a marker of clusters 0 and 2, has been shown by us and others to mark a pancreas duct cell type enriched in progenitor capacity (*Qadir et al., 2020*; *Kilic et al., 2006*). *Anxa3*, a marker of clusters 1 and 4, inhibits phospholipase A2 and cleaves inositol 1,2-cyclic phosphate-generating inositol 1-phosphate in a calcium-dependent manner (*Tait et al., 1993*; *Gerke and Moss, 2002*). *Gmnn* expression is highly conserved and plays crucial roles in development biology, yet its function in normal pancreatic duct cells is incompletely understood (*Kushwaha et al., 2016*). *Gmnn*, a marker of cluster 0, acts to inhibit re-replication of DNA during DNA synthesis by inhibiting the prereplication complex (*McGarry and Kirschner, 1998*; *Ballabeni et al., 2013*). Understanding the function of a gene in normal physiology is central to dissecting its role in disease. To get a better understanding of the function of these subpopulation markers in normal human pancreatic duct cells, we next examined the consequences of their loss in the immortalized human pancreatic duct cell line HPDE6c7 (*Ouyang et al., 2000*). HPDE6c7 cells demonstrate several features of normal pancreatic duct epithelium including gene expression of *MUC1*, *CA2*, and *KRT19* and have been used in many investigations as an *in vitro* model of 'near normal' human pancreatic duct cells (*Ouyang et al., 2000*; *Furukawa et al., 1996*; *Qian et al., 2005*; *Lee et al., 2017*). We generated and validated *SPP1*, *GMNN*, and *ANXA3* knockout (KO) HPDE6c7 lines using CRISPR/Cas9 (*Figure 6A–C*). Strong, consistent phenotypes were observed among different KO lines for each gene despite some lines not demonstrating full loss of the protein (HPDE6c7 *ANXA3* gRNA2 and HPDE6c7 *SPP1* gRNAs 1–4). Cellular morphology was similar to the scrambled (scr) gRNA control (*Suzuki et al., 2016*) for every KO line except HPDE6c7 *SPP1* gRNAs 1–4, which displayed a dramatic change in cellular morphology. HPDE6c7 *SPP1* KO cells showed prominent filipodia and significantly increased proliferation when compared to the HPDE6c7 scr gRNA control, a phenotype suggestive of increased progenitor function (*Figure 6D, E*). The change in cellular morphology in HPDE6c7 *SPP1* KO lines is accompanied by decreased duct function as measured by carbonic anhydrase activity (*Figure 6F*).

To assess the changes in HPDE6c7 *SPP1* KO lines on a molecular scale, we performed bulk RNA-sequencing on all four HPDE6c7 *SPP1* KO lines and the HPDE6c7 scr gRNA control. A significant increase in markers associated with EMT (*VIM*, *ZEB1*, *TWIST1*, *MMP2*) was observed in HPDE6c7

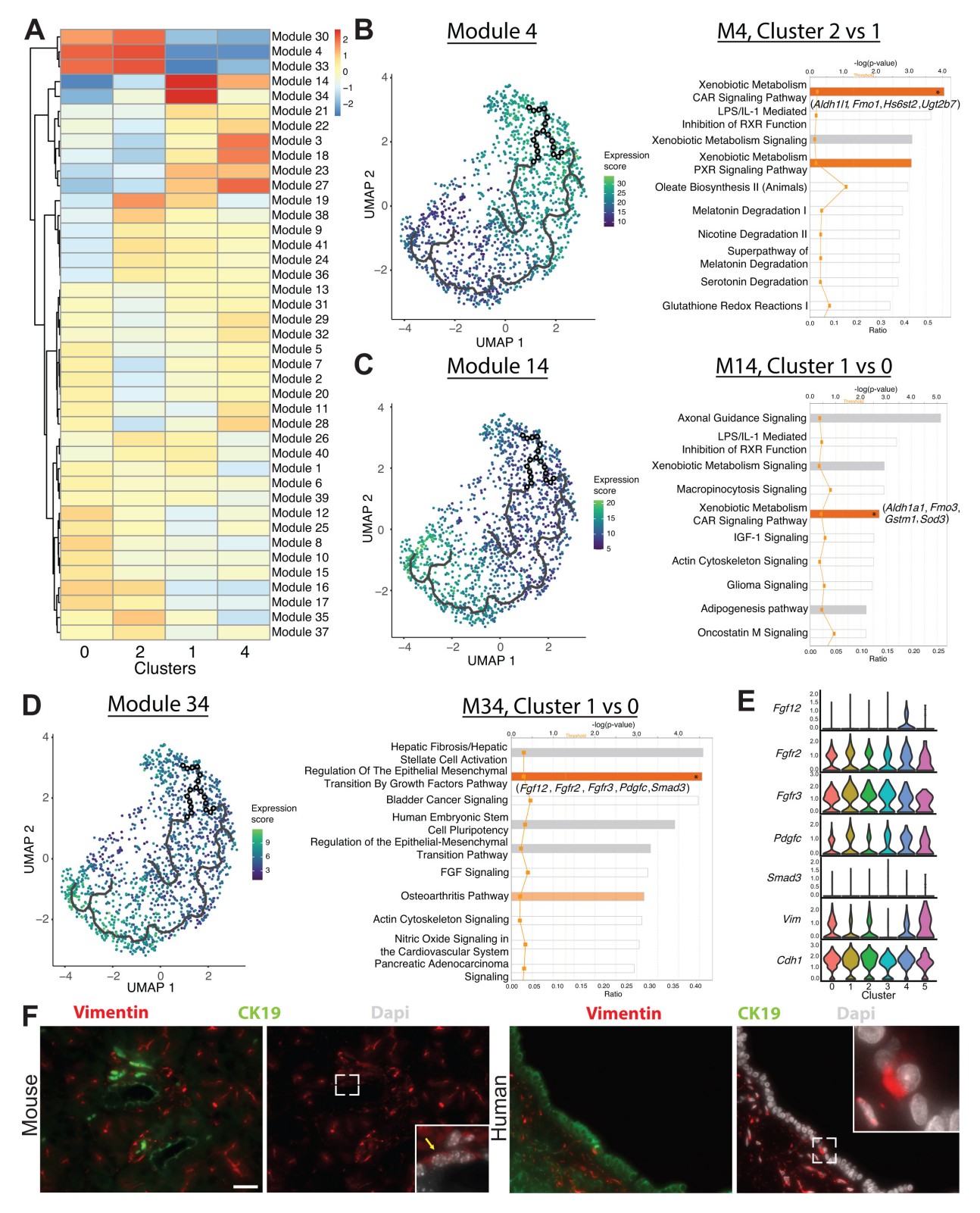

**Figure 5.** Monocle 3 analysis reveals an epithelial-mesenchymal axis in pancreatic duct cells. (A) Expression changes of the modules generated by Monocle 3 analysis are shown for each cluster. (B–D) Expression of modules 4, 14, and 34 along with select ingenuity pathways analysis results of the top 10 deregulated pathways are shown. Genes in parenthesis are altered in the pathway containing an asterisk in the bar. (E) Stacked violin plots show expression of genes in the regulation of the epithelial-mesenchymal transition by growth factors pathway in DBA[+] duct clusters 0–5. (F)

*Figure 5 continued on next page*

*Figure 5 continued*

Immunofluorescence depicts CK19$^+$ vimentin$^+$ copositive pancreatic duct cells in mouse (yellow arrow) and human. The main pancreatic duct is shown for humans. Scale bars are 50 µM.

The online version of this article includes the following source data and figure supplement(s) for figure 5:

**Source data 1.** Ingenuity pathways analysis results comparing select modules in Monocle 3 analysis.
**Source data 2.** Ingenuity pathways analysis upstream regulator analysis comparing select modules in Monocle 3 analysis.
**Source data 3.** Log fold change analysis comparing select modules in Monocle 3 analysis.
**Figure supplement 1.** RaceID3 clusters and Monocle 3 analysis.

SPP1 KO lines when compared to the control (*Figure 6—source data 1–4*, *Figure 6G–H*, and *Figure 6—figure supplement 1A–C*). Markers of mature duct cells (*HNF1B*, *SOX9*, *KRT19*) were significantly downregulated in HPDE6c7 SPP1 KO lines when compared to the control (*Figure 6G, I, J* and *Figure 6—source data 1*). Gene set enrichment analysis (GSEA) suggested positive enrichment of pathways that regulate embryogenesis (Hox genes and Notch signaling) and cell cycle regulation in HPDE6c7 SPP1 KO lines when compared to the HPDE6c7 scr gRNA control (*Figure 6—figure supplement 1D–F*). Additionally, qPCR analysis demonstrated a significant increase in pancreatic progenitor markers (*Gu et al., 2004*; *Willmann et al., 2016*) in HPDE6c7 SPP1 KO lines when compared to the HPDE6c7 scr gRNA control, supporting the notion that loss of OPN leads to a more immature, progenitor-like state (*Figure 6K*). Taken together, these results define unique functional properties for markers that characterize murine DBA$^+$ pancreas duct cells and suggest that SPP1 is an essential regulator of human pancreatic duct cell maturation and function.

Transdifferentiation of pancreatic duct cells to endocrine cells at early postnatal stages and in pancreatic injury models has been suggested by several studies (*Bonner-Weir et al., 2008*; *Kim and Lee, 2016*). To query whether HPDE6c7 SPP KO progenitor-like, dedifferentiated duct cells harbor the capacity to redifferentiate to endocrine cells *in vivo*, we injected HPDE6c7 SPP1 KO cell lines and HPDE6c7 scr gRNA control cells subcutaneously into NSG mice (*Figure 7A*). After 5 days post-injection, α-amylase$^+$ CK19$^+$ double-positive cells were evident in HPDE6c7 scr gRNA control cells, but not in HPDE6c7 SPP1 KO cells (*Figure 7B*). This observation is consistent with the previously described ducto-acinar axis characteristic of human pancreatic duct cells (*Qadir et al., 2020*). We observed Neurogenin-3$^+$SOX9$^+$ copositive HPDE6c7 SPP1 KO cells, suggesting potential for differentiation towards the endocrine lineage (*Figure 7C*). We detected Synaptophysin$^+$ Glucagon$^+$ as well as Synaptophysin$^+$ C-peptide$^+$ double-positive HPDE6c7 SPP1 KO cells (*Figure 7D, E*). Expression of endocrine markers in subcutaneously injected HPDE6c7 scr gRNA cells was not observed at day 5 post-injection. A small subset of C-peptide$^+$ HPDE6c7 SPP1 KO cells express both PDX1 and NKX6.1 (*Figure 7E, F*). Together, these data point to previously unappreciated roles for SPP1 in maintaining duct cell properties and preventing changes in cell identity.

## Geminin safeguards against accumulation of DNA damage in mouse ductal cells in the setting of CP

One marker of the workhorse population of pancreatic duct cells *Gmnn* has previously been associated with chronic inflammatory diseases such as asthma (*Garbacki et al., 2011*). We therefore queried its role in pancreas inflammatory disease. *Gmnn* binds to CDT1 and inhibits DNA replication during the S phase. Geminin is a crucial regulator of genomic stability; its inhibition in multiple cancer cell lines leads to DNA re-replication and aneuploidy (*Zhu and Depamphilis, 2009*; *Saxena and Dutta, 2005*). To determine the requirement for *Gmnn* in normal homeostatic pancreatic duct cells, we generated a conditional *Gmnn* floxed allele and crossed the mouse to the *Sox9-CreERT2* (*Kopp et al., 2011*) and *Hnf1b*$^{CreERT2}$ (*Solar et al., 2009*) transgenic lines (*Figure 8—figure supplement 1*). Adult mice, between the ages of 7–9 weeks, were injected with tamoxifen to ablate Geminin in mouse pancreatic duct cells. Tamoxifen-injected *Sox9-CreERT2*$^{Tg/wt}$; *Geminin*$^{f/f}$, *Sox9-CreERT2*$^{Tg/wt}$; *Geminin*$^{f/wt}$, and *Hnf1b*$^{CreERT2\ Tg/wt}$; *Geminin*$^{f/f}$ mice displayed no histological abnormalities as assessed by hematoxylin and eosin (H&E) staining and no significant alterations in DNA damage as assessed by ATR and γ-H2AX IF up to 6 months post-tamoxifen injection (data not shown). We were unsurprised by these findings, given the low proliferation rate of murine pancreatic duct cells suggested by our single-cell data. Thus, Geminin may only be required in the context of

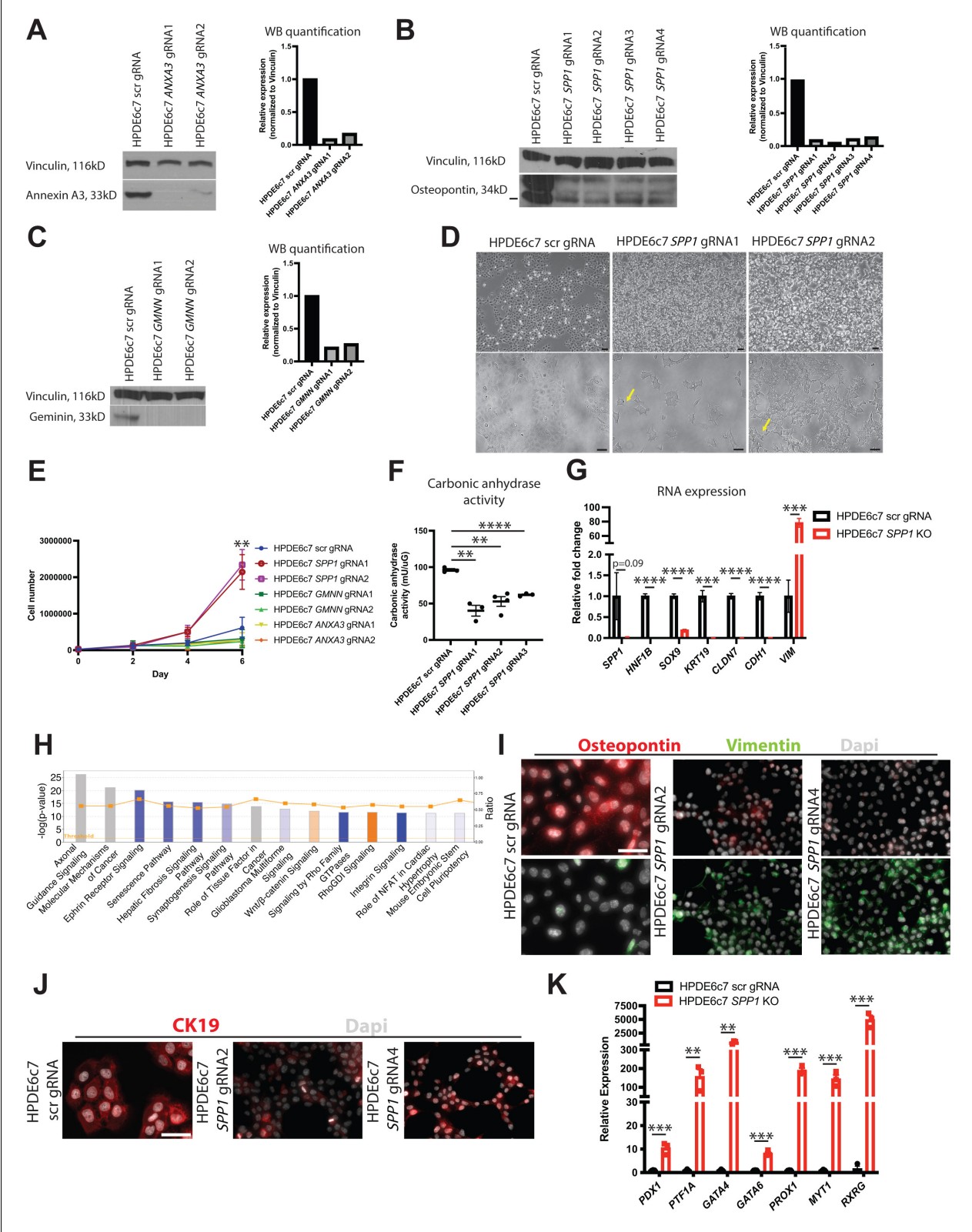

**Figure 6.** *SPP1* loss promotes a progenitor-like state in human pancreatic duct cells. (**A–C**) Western blot and quantification of western blot images shows expression of Annexin A3, osteopontin, and geminin in knockout (KO) HPDE6c7 lines and the control. (**D**) Brightfield images show changes in cellular morphology of HPDE6c7 *SPP1* KO lines. Yellow arrows point to filipodia. Scale bars are 100 μm. (**E**) Cell counting demonstrates a significant increase in cell number at day 6 in HPDE6c7 *SPP1* KO cells when compared to the HPDE6c7 scrambled (scr) gRNA control (p=0.0089 for HPDE6c7 *SPP1*
*Figure 6 continued on next page*

*Figure 6 continued*

gRNA1 and p=0.0042 for HPDE6c7 *SPP1* gRNA2). (F) Significantly decreased carbonic anhydrase activity is observed in HPDE6c7 *SPP1* KO lines when compared to the control. (G) Relative fold changes calculated using rpm values of mesenchymal and duct markers are shown. Average counts (normalized to library size) for *SPP1* are 84.0 ± 47.3 (scr) and 2.1 ± 0.4 (KO), *HNF1B* are 434.9 ± 24.8 (scr) and 0.5 ± 0.3 (KO), *SOX9* are 1,617.0 ± 87.2 (scr) and 302.4 ± 35.3 (KO), *KRT19* are 17,458.1 ± 2,367.2 (scr) and 60.7 ± 60.7 (KO), *CLDN7* are 3,381.1 ± 222.9 (scr) and 17.6 ± 3.7 (KO), *CDH1* are 8,995.8 ± 805.9 (scr) and 109.1 ± 29.0 (KO), and *VIM* are 84.1 ± 32.4 (scr) and 6,577.2 ± 513.7 (KO). (H) The top 14 deregulated pathways from ingenuity pathways analysis are shown comparing HPDE6c7 *SPP1* KO vs. HPDE6c7 scr gRNA control. (I) Immunocytochemistry (ICC) demonstrated reduced osteopontin expression in HPDE6c7 gRNA2 and HPDE6c7 *SPP1* gRNA4 when compared to HPDE6c7 scr gRNA. Vimentin ICC depicts organized intermediate filaments in HPDE6c7 *SPP1* gRNA2 and HPDE6c7 *SPP1* gRNA4 while HPDE6c7 scr gRNA cells show diffuse, light labeling. Scale bar is 50 μm. (J) CK19 ICC shows organized intermediate filaments in HPDE6c7 scr gRNA cells while HPDE6c7 *SPP1* gRNA2 and HPDE6c7 *SPP1* gRNA4 cells display punctate CK19 labeling, where present. Scale bar denotes 50 μm. (K) qPCR results of pancreatic progenitor markers are shown for HPDE6c7 scr gRNA control and HPDE6c7 *SPP1* KO lines.

The online version of this article includes the following source data and figure supplement(s) for figure 6:

**Source data 1.** Differentially expressed genes comparing HPDE6c7 *SPP1* knockout vs. HPDE6c7 scrambled gRNA.
**Source data 2.** Ingenuity pathways analysis results comparing HPDE6c7 *SPP1* knockout vs. HPDE6c7 scrambled gRNA.
**Source data 3.** Ingenuity pathways analysis upstream regulator analysis comparing HPDE6c7 *SPP1* knockout vs. HPDE6c7 scrambled gRNA.
**Source data 4.** Filtered normalized (to library size) counts for RNA-seq of HPDE6c7 scr gRNA and HPDE6c7 *SPP1* knockout cell lines.
**Figure supplement 1.** Characterization of DBA[+] murine ductal markers.

pathologies characterized by increased proliferation in the pancreas such as pancreatitis or PDA (*Salabat et al., 2008*).

We examined proliferation in human pancreas duct cells in CP patients (N = 5 patients) and found a significant increase in geminin expression when compared to normal human pancreatic duct cells (N = 10 donors) (*Figure 8A, B*). Pancreatic duct ligation (PDL), an experimental technique that recapitulates features of human gallstone pancreatitis, results in an increase in proliferation of rat pancreatic duct cells (*Githens, 1988*; *Walker and Pound, 1983*). To investigate the role of Geminin in mouse pancreatic duct cells in the setting of CP, we performed PDL on *Sox9-CreERT2^{Tg/wt}*; *Geminin^{f/f}*, *Sox9-CreERT2^{Tg/wt}*; *Geminin^{f/wt}*, *Hnf1b^{CreERT2 Tg/wt}*; *Geminin^{f/f}* and littermate control mice (*Figure 8C*). As in the human setting, we also observed upregulation of Geminin in ductal epithelium in the control PDL mouse group (*Figure 8D*). Previously reported features of the PDL model were evident in our transgenic mice including replacement of parenchymal cells with adipose tissue, inflammation, and fibrosis (*Aghdassi et al., 2011*; *Rastellini et al., 2015*; *Figure 8—figure supplement 2A, B*). Significant attenuation of Geminin expression was observed in *Sox9-CreERT2^{Tg/wt}*; *Geminin^{f/f}*, *Sox9-CreERT2^{Tg/wt}*; *Geminin^{f/wt}*, and *Hnf1b^{CreERT2 Tg/wt}*; *Geminin^{f/f}* mouse pancreatic duct cells when compared to controls (*Figure 8D* and *Figure 8—figure supplement 3A*). Homozygous *Gmnn* loss in SOX9[+] pancreatic ductal cells promoted an acute increase in proliferation, as assessed by BrdU incorporation, at day 7, which became insignificant at day 30 (*Figure 8—figure supplement 3B-E*). No changes were observed in apoptosis for any model or time point when compared to controls as assessed by cleaved caspase-3 IF (data not shown). Examination of duct cell DNA damage by γ-H2AX IF showed significantly increased γ-H2AX foci in *Sox9-CreERT2^{Tg/wt}*; *Geminin^{f/f}* mice at day 7, an observation that was sustained at day 30 (*Figure 8E–H*). Assessment of DNA damage in *Sox9-CreERT2^{Tg/wt}*; *Geminin^{f/f}*, *Sox9-CreERT2^{Tg/wt}*; *Geminin^{f/wt}*, and *Hnf1b^{CreERT2 Tg/wt}*; *Geminin^{f/f}* mice by ATR IF showed no significant changes (data not shown). The lack of phenotypes observed in the *Hnf1b^{CreERT2 Tg/wt}*; *Geminin^{f/f}* model may be due to differences in recombination induced by the *Sox9-CreERT2* and *Hnf1b^{CreERT2}* lines since fewer pancreatic duct cells express HNF1B (*Figure 1—figure supplement 1G* and *Figure 8C*). Taken together, these data suggest that Geminin is an important regulator of genomic stability in pancreatic ducta cells in the setting of CP.

## Discussion

We present a single-cell transcriptional blueprint of murine pancreatic duct cells, intrapancreatic bile duct cells, and pancreatobiliary cells. Notably, our single-cell analysis indicated that endothelial cells, fibroblasts, and immune cells are also obtained using the DBA[+] lectin sorting strategy (*Reichert et al., 2013*) and suggests that a subsequent ductal purification step is required to obtain

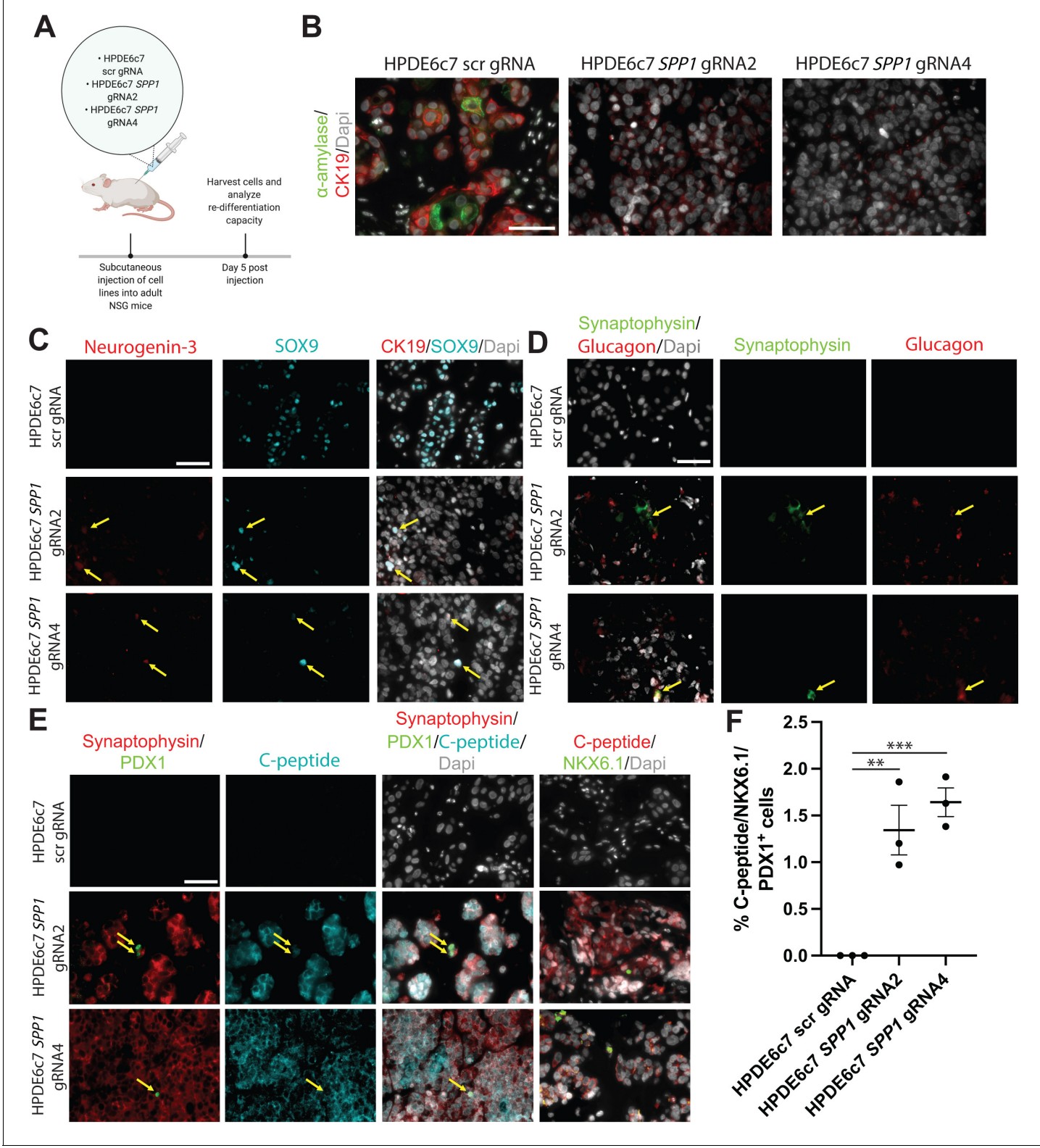

**Figure 7.** HPDE *SPP1* knockout (KO) cells are capable of differentiating into cells with endocrine appearance, including cells exhibiting α- and β-like appearance, but not duct-like or acinar-like cells *in vivo*. (A) Schematic of *in vivo* experiment. (B) Immunofluorescence (IF) shows CK19+ α-amylase+ double-positive HPDE6c7 scrambled (scr) gRNA cells. (C) IF depicts NGN3+ SOX9+ double-positive HPDE6c7 *SPP1* KO cells (yellow arrows). (D) Synaptophysin+ Glucagon+ double-positive cells (yellow arrows) are detected in HPDE6c7 *SPP1* KO cells. (E) C-peptide, synaptophysin, NKX6.1, and
*Figure 7 continued on next page*

Figure 7 continued

PDX1 expression are evident in HPDE6c7 *SPP1* KO cells. C-peptide, synaptophysin, and PDX1 triple-positive cells are highlighted with yellow arrows. (F) The percentage of C-peptide[+], NKX6.1[+], and PDX1[+] triple-positive cells for HPDE6c7 scr gRNA cells is 0, HPDE6c7 *SPP1* gRNA2 is 1.343 ± 0.266, and HPDE6c7 *SPP1* gRNA 4 cells is 1.642 ± 0.153. All scale bars in this figure are 50 μm.

pure pancreatic duct cells using this protocol. A static transcriptional picture in time has highlighted a very dynamic view of pancreas duct cell heterogeneity. Our study provokes reinterpretation of several previously published lineage tracing reports using ductal-specific Cre mouse lines and will help plan future lineage tracing studies.

Cluster 0 workhorse pancreatic duct cells comprise the largest pancreatic duct subpopulation identified. Although clusters 0 and 2 share many markers, we found compelling differences in metabolic states as manifested in part by an overall lower gene and transcript count for cluster 2. IPA suggested that subpopulations of pancreatic duct cells may use different predominant mechanisms for bicarbonate secretion such as CFTR (*Ishiguro et al., 2009*) for cluster 0 and calcium signaling for cluster 1 (*Ishiguro et al., 2012*). One notable difference between clusters 0 and 2 vs. 1 is the molecules that regulate the xenobiotic metabolism CAR signaling pathway. We observed expression of several genes, whose alteration contributes to PDA progression including *Tgfb2* and *Ctnnb1* in cluster 0 and *Ppp1r1b*, *Smarca4*, and *Tgfb1* in cluster 1. IPA upstream regulator analysis of Monocle 3 module 14 predicted significant inhibition of *Kras* in cluster 1 when compared to cluster 0. Additionally, IPA upstream regulator analysis comparing cluster 2 vs. 0 in module 19 predicted activation of *Myc* and *Mycn* in cluster 2. These genes play central roles in homeostasis of pancreatic duct cells, and it is possible that distinct ductal cell subpopulations that are actively expressing these pathways may have different predispositions to PDA with mutations in these genes, heterogeneity which may also contribute to the development of different subtypes of PDA.

The role of *Spp1* in homeostatic pancreatic duct cells has been elusive since *Spp1* KO mice have no apparent pancreatic duct phenotypes (*Kilic et al., 2006*). We identified an EMT axis in pancreatic duct cells using Monocle 3 and validated this observation in mouse and human duct cells. *Spp1* is one gatekeeper of this epithelial to mesenchymal transitory duct phenotype as manifested by loss of ductal markers, reduced duct function, and upregulation of EMT genes in HPDE6c7 *SPP1* KO cells when compared to controls. Clusters 0 and 2, characterized by strong expression of *Spp1*, show the highest StemID2 scores. *SPP1* KO HPDE6c7 cells display prominent filipodia and the highest proliferative capacity of all markers examined when compared to the control. Taken together, these phenotypes, along with upregulation of pathways regulating mammalian development (Notch signaling and Hox genes) manifested by GSEA, suggest that *SPP1* loss promotes human duct cell dedifferentiation.

During pancreas development, the multipotent epithelial progenitors become increasingly compartmentalized into tip and trunk progenitors that give rise to acinar and endocrine/ductal cells, respectively (*Zhou et al., 2007*). Our data suggest that OPN-deficient HPDE6c7 cells dedifferentiate into a trunk, and not tip, progenitor-like cell and that redifferentiation of HPDE6c7 *SPP1* KO cells to a human pancreatic duct or acinar cell lineage is not favored *in vivo.* These data underscore the requirement for OPN expression for mature human pancreas duct cell identity. It has been hypothesized that *SPP1*'s role in mature pancreatic duct cells is evident during pathogenesis. Several groups have already nicely shown that *Spp1* plays important roles in pancreatic pathologies including PDA (*Adams et al., 2019*; *Zhao et al., 2018*). In human pancreas duct cells, the subpopulation characterized by *SPP1* expression is described as 'stress/harboring progenitor-like cells' (*Qadir et al., 2020*). We observed significant deregulation of 14 cancer-related IPA pathways for which pathway directionality was known in HPDE6c7 *SPP1* KO lines vs. HPDE6c7 scr gRNA controls. 13/14 of these cancer-related pathways, including pancreatic adenocarcinoma signaling, were in a direction suggestive that *SPP1* loss protects against tumor progression in human pancreatic duct cells. These findings are in agreement with published studies suggesting that *SPP1* loss ameliorates aggressiveness of pancreatic cancer cells (*Kolb et al., 2005*; *Adams et al., 2019*) and colon cancer cells (*Zhao et al., 2018*; *Ishigamori et al., 2017*).

The requirement for Geminin in the prevention of DNA re-replication initiation has been postulated to be when cells are stressed to divide quickly (*Barry et al., 2012*). We were unable to detect DNA damage with Geminin loss in homeostatic pancreatic duct cells, which may be due to the low

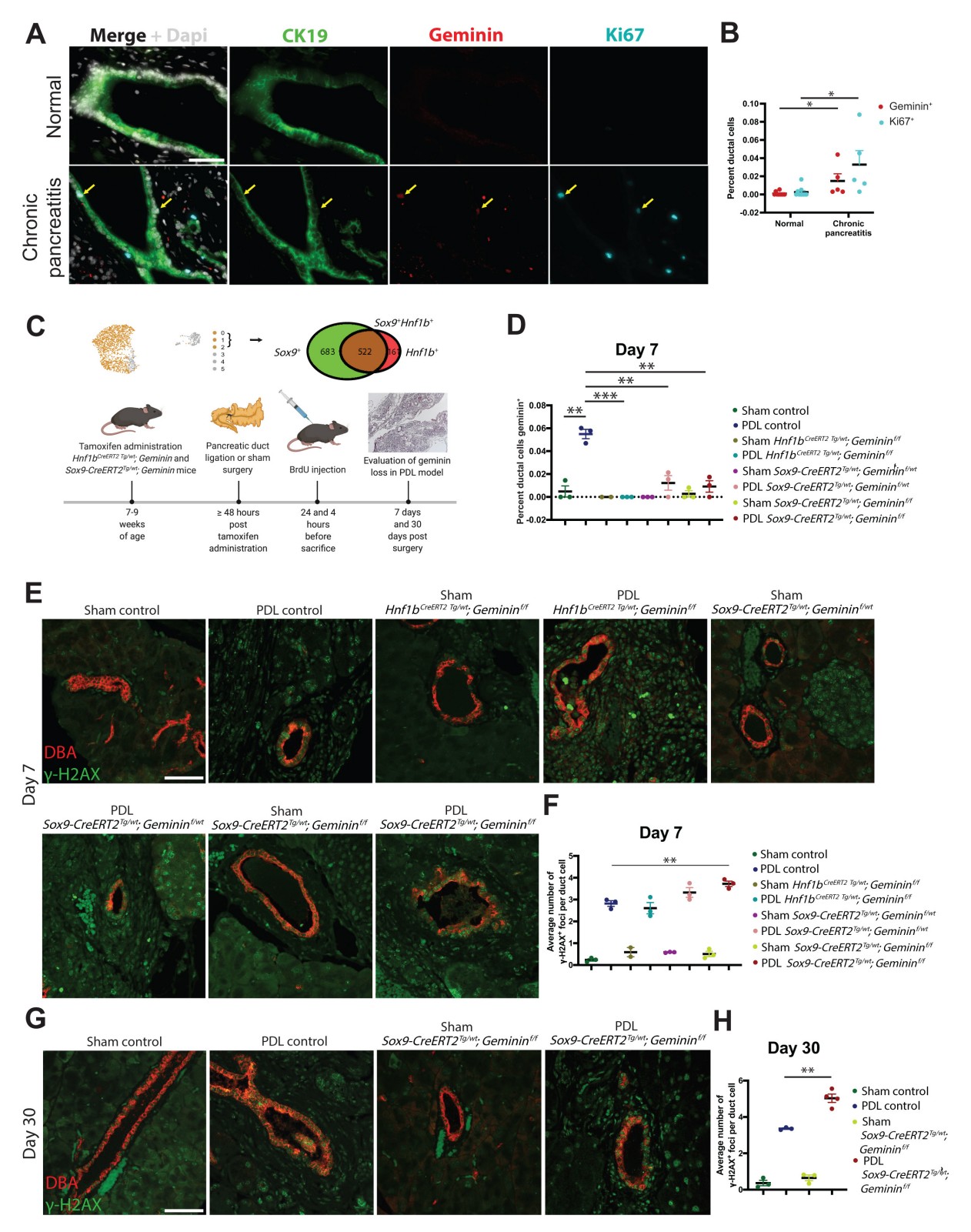

**Figure 8.** Geminin is a regulator of genomic stability in mouse pancreatic duct cells during chronic pancreatitis (CP). (A, B) High-magnification immunofluorescence (IF)images and quantification show a significant increase in proliferation in pancreatic duct cells in CP patients when compared to normal human pancreatic duct cells. Yellow arrows point to Geminin$^+$ Ki67$^+$ copositive cells. (C) A schematic of tamoxifen and BrdU administration is shown. The Uniform Manifold Approximation and Projection depicts the pancreatic cells (clusters 0–2) that were analyzed in this experiment. The Venn

*Figure 8 continued on next page*

Figure 8 continued

diagram shows the number of cells in clusters 0–2 that are SOX9$^+$, HNF1B$^+$, and SOX9$^+$HNF1B$^+$ copositive. (D) Quantification of Geminin-positive ductal cells at day 7 in *Sox9-CreERT2$^{Tg/wt}$; Geminin$^{f/f}$, Sox9-CreERT2$^{Tg/wt}$; Geminin$^{f/wt}$, Hnf1b$^{CreERT2\ Tg/wt}$; Geminin$^{f/f}$* and control mice is depicted. (E, F) Representative IF images and quantification of γ-H2AX-positive foci are shown at day 7 in the pancreatic duct ligation (PDL) transgenic models. (G, H) Representative IF images and quantification of γ-H2AX-positive foci are shown at day 30 in the *Sox9-CreERT2$^{Tg/wt}$; Geminin$^{f/f}$* and control PDL models. All scale bars in this figure are 50 μm.

The online version of this article includes the following figure supplement(s) for figure 8:

**Figure supplement 1.** Generation of 2loxP and 1loxP heterozygous ES cells for mouse *Geminin.*
**Figure supplement 2.** Histology of transgenic pancreatic duct ligation (PDL) models.
**Figure supplement 3.** Geminin loss causes a transient proliferation response in *Sox9-CreERT2$^{Tg/wt}$; Geminin$^{f/f}$* mice.

proliferation rate of pancreatic duct cells and/or the presence of compensatory mechanisms with redundant function, such as ubiquitin-dependent degradation of CDT1 at the time of replication licensing (*Arias and Walter, 2005*; *Kerns et al., 2007*; *Li and Blow, 2005*; *Maiorano et al., 2005*). Compensatory mechanisms are not sufficient to rescue the effects of Geminin loss in pancreatic duct cells in the context of CP, the result of which is accumulation of sustained DNA damage evident by γ-H2AX, but not ATR labeling. It has been previously reported that ATR is activated in Geminin-depleted colon cancer cell lines (*Lin and Dutta, 2007*). Activation of the ATR-CHK1 pathway is not a major player in pancreatic duct cells in the setting of CP (*Smith et al., 2010*), suggesting that different mechanisms participate in sensing Geminin depletion-induced DNA damage in different experimental systems and tissues. While our limited functional analyses of the *SPP1* and *Gmnn* mutant models provide important information regarding their function in pancreas duct cells, additional studies will be required to fully understand their roles in normal and disease pancreatic physiology.

## Materials and methods

### Preparation of pancreatic duct cells for single-cell analysis

Pancreata from four 9-week-old female C57BL/6J littermates (Jackson Labs, Stock 000664) were dissected, digested into single cells, and the DBA$^+$ fraction obtained as previously described (*Reichert et al., 2013*). Subsequently, live DBA$^+$ cells were isolated for scRNA-seq by excluding propidium iodide (Thermo Fisher Scientific, P3566)-positive single cells during FACS. scRNA-seq was performed by the Institute for Human Genetics Genomics Core Facility at University of California San Francisco (UCSF) using the 10X Genomics platform. Briefly, live, single, DBA$^+$ pancreatic cells were loaded onto the microfluidic chip to generate single-cell GEMs (Gel Bead-In-Emulsions). Following cell lysis and unique barcode labeling, the cDNA library of 18,624 live pancreatic cells was generated using the Chromium Single Cell 3' GEM, Library and Gel Bead Kit v2 (10X Genomics). The cDNA library was sequenced on one lane using an Illumina HiSeq 4000.

scRNA-seq data was generated on the 10X platform (10X Genomics, Pleasanton, CA) according to Single Cell 3' protocol (v2 Chemistry) recommended by the manufacturer (*Zheng et al., 2017*). The Cell Ranger software pipeline (version 2.1.1) was used to demultiplex cellular barcodes, map reads to the genome and transcriptome using the STAR aligner, and produce a matrix of gene counts versus cells. Doublets were filtered by excluding cells having RNA counts > 30,000 and mitochondrial genes percentage >10% in addition to using Scrublet (*Wolock et al., 2019*). The R package Seurat (*Satija et al., 2015*) was used to process the unique molecular identifier count matrix and perform data normalization (gene expression measurements for each cell were normalized by total expression, and log-transformed), dimensionality reduction, clustering, duct cell isolation, and differential expression analyses. We identified three clusters enriched in genes from two different cell types including (1) acinar and T cell, (2) acinar and macrophage, and (3) acinar and duct cell. Because our dataset does not contain a population of acinar cells (they are not DBA$^+$), doublet detector algorithms will not remove acinar cell doublets from our dataset. Based on this reasoning, we removed these clusters containing a high threshold level of expression of acinar cell genes.

## Generation of *Gmnn* conditional floxed allele

The general strategy to achieve Cre recombinase-mediated conditional gene ablation was to flank exons 3 and 4 of *Mus musculus Gmnn* by loxP sites (*Figure 8—figure supplement 1A*). The arms of homology for the targeting construct were amplified from BAC clone RP23-92G13 by PCR with high-fidelity Taq polymerase. One primer contained a loxP site and a single SphI site, which was used to verify the presence of the loxP site associated with it. Finally, the selectable cassette *CMV-hygro-TK* was incorporated into the targeting vector. The selectable marker itself was flanked by two additional loxP sites generating a targeting vector containing three loxP sites. Such a strategy allows the generation of ES cells with both a KO allele and a conditional KO allele after Cre-mediated removal of the selection cassette *in vitro*. The targeting vector was sequenced to guarantee sequence fidelity of exons 3–4 and the proper unidirectional orientation of the three loxP sites. The complete left arm of homology was about 3,200 bp in length, and the right arm of homology was 2,100 bp in length.

V6.5 ES cells were electroporated (25 µF, 400V) with the three loxP sites-containing targeting construct, and hygromycin selection was performed to identify correctly targeted ES cells. Successfully targeted ES cells (3loxP) were identified with Southern blot (*Figure 8—figure supplement 1B*). These 3loxP ES cells were then electroporated with a Cre-expressing plasmid and counter-selected with ganciclovir. ES cells that contained either one loxP or two loxP sites, respectively, were identified by Southern blot (*Figure 8—figure supplement 1C*). An ES cell clone was chosen that carried the conditional KO allele (two loxP sites flanking exons 3 and 4) and was used for blastocyst injections to generate chimeric founder mice. *Gmnn^{f/f}* mice displayed normal litter sizes. For routine genotyping of *Gmnn^{f/f}* mice, the primers GCCTCGAACTCAGAAATCCA (primer A) and AACACAAAATTTGGCCTGCT (primer B) were used. To identify the deleted allele by PCR, primer C (TAGCCCGGACTACACAGAGG) can be used with primer A.

## Southern blot

For Southern blotting of genomic DNA, samples were digested with SphI or Bsu36I restriction enzymes for at least 4 hours and separated on an 0.8% agarose gel. The DNA was transferred to a Hybond-XL membrane (GE-Healthcare) in a custom transfer setup. Before assembly, the agarose gel was treated for 15 min in depurination solution (21.5 ml 37% HCl in 1L $H_2O$), briefly rinsed in $H_2O$, and then soaked in denaturing solution (20 g NaOH pellets, 87.6 g NaCl in 1 l $H_2O$) for 30 min. After transfer, the DNA was crosslinked to the membrane with UV light. The PCR-amplified external Southern blot probes were labeled with $^{32}P$ using the Prime-It II Random Primer Labeling kit from Stratagene. After hybridization of the probe and washing of the membrane, Kodak MS film was exposed to it and subsequently developed.

## Mice

NSG mice from Jackson Labs (Stock 005557) were used. The transgenic mouse strain *Sox9-CreERT2* was obtained from Jackson Labs (Stock 018829), and *Hnf1b^{CreERT2}* has been previously described (*Solar et al., 2009*). Mice were maintained on a mixed genetic background. To induce Cre recombination, mice were injected with 6.7 mg tamoxifen (Actavis, NDC 0591-2473-30) via oral gavage three different days over the course of a week at 7–9 weeks of age. PDLs were performed as previously described (*De Groef, 2015*). BrdU (Sigma, B9285-1G) injections were performed 24 hours and 4 hours prior to dissection. 100,000 HPDE6c7 cells mixed 1:1 with media containing cells and growth factor reduced matrigel (Corning, 356231) in a total volume of 100 µl were injected subcutaneously into NSG mice. Mice were genotyped by PCR or Transnetyx. All animal studies were approved by the Institutional Animal Care and Use Committee at UCSF.

## Histology/immunostaining

Tissues were fixed in Z-Fix (Anatech Ltd., 174), processed according to a standard protocol, and embedded in Paraplast Plus embedding agent for histology, with DMSO (VWR 15159–464).

For immunostaining, paraffin sections were deparaffinized, rehydrated, and antigen retrieval was performed, for all antibodies except BrdU, with Antigen Retrieval Citra (Biogenex, HK086-9K) using a heat-mediated microwave method. For immunostaining of BrdU, antigen retrieval was performed as previously described (*Puri et al., 2018*). For immunohistochemistry (IHC), endogenous peroxidase

activity was blocked by incubation with 3% hydrogen peroxide (Fisher Scientific, H325-100) prior to antigen retrieval. Primary antibodies were incubated overnight at 4℃. Secondary antibodies were used at 4 µg/ml and incubated at room temperature (RT) for 1 hour (IHC) or 2 hours (IF). For IF, slides were mounted in ProLong Diamond Antifade Mountant with DAPI (Thermo Fisher, P36962). For IHC, Vectastain Elite ABC kit (Vector Laboratories, PK-6100) and DAB Peroxidase (HRP) Substrate kit (Vector Laboratories, SK-4100) were used. Primary antibodies used in this study are listed in *Supplementary file 2*. Secondary antibodies used in this study were obtained from Invitrogen, Jackson ImmunoResearch, and Biotium.

Immunostaining of cluster markers as well as the types of ducts within the ductal hierarchy tree were reviewed and classified by a board-certified pathologist. For expression analysis of selected markers in murine and human tissues, images shown are representative of at least three different donors or 9-week-old C57BL/6J mice. For quantification of BrdU, cleaved caspase-3, Geminin, Ki67, γ-H2AX, and ATR, at least 60 cells from three different ducts were analyzed. Quantification of γ-H2AX foci included duct cells with zero foci. For quantification of c-peptide/PDX1/NKX6.1 triple-positive cells, at least three images containing an average of 188 cells were counted from three technical replicates per cell line. Normal human tissue used in this study was obtained from research-consented human cadaver donors through UCSF's Islet Production Core. Human pancreatic tissue specimens from five surgical resections from patients without pancreaticobiliary carcinoma or high-grade pancreatic intraepithelial neoplasia were obtained. The pancreatic histologic section demonstrated CP with loss of acinar parenchyma, resulting in atrophic lobules along with variable fibrosis and chronic inflammation (most had no to sparse lymphocytic inflammation).

## Immunocytochemistry

Cells were grown on coverslips in six-well plates and fixed at RT for 15 min with 4% paraformaldehyde. Cells were permeabilized with permeabilization solution (0.1% w/v Saponin, 5% w/v BSA in PBS-/-). The primary antibody was incubated in permeabilization solution at 4℃ overnight. After washing off unbound primary antibody with PBS-/-, the secondary antibody was incubated in permeabilization solution for 1 hour at RT. After washing off unbound secondary antibody with PBS-/-, cells were mounted using ProLong Diamond Antifade Mountant with DAPI (Thermo Fisher, P36962).

## Flow cytometry

For analysis of cell surface markers (EPCAM), cells were resuspended in FACS buffer (1% FBS + 2 mM EDTA in PBS -Mg/-Ca), filtered in FACS tubes with a cell strainer cap, and spun at 1350 rpm for 3–5 min. The supernatant was discarded, and cells were resuspended in 100 µl of directly conjugated primary antibody diluted in FACS buffer and stained for 60 min at RT. Stained cells were washed with 2 ml FACS buffer and spun at 1350 rpm for 3–5 min. The supernatant was discarded and cells were resuspended in 250 µl FACS buffer containing 0.5 µg/ml DAPI immediately before analyzing on the flow cytometer.

For analysis of intracellular antigens, single-cell suspensions of cell lines were prepared. Cells were washed with PBS -Mg/-Ca, resuspended in 250 µl FACS buffer, and filtered in FACS tubes with a cell strainer cap. 2 ml 1X permeabilization buffer (Affymetrix eBiosciences, 00-8333-56) was added to cells, and cells were subsequently spun at 1500 rpm for 5 min. Supernatant was removed, and 100 µl primary antibody diluted in CAS-Block (Invitrogen, 00–8120) + 0.2% TritonX-100 was added to cells. Cells were stained overnight at 4℃. Subsequently, cells were washed with 3 ml 1X permeabilization buffer and spun at 1500 rpm for 5 min. The supernatant was discarded. If using directly conjugated primary antibodies, cells were resuspended in 250 µl FACS buffer and analyzed on the flow cytometer. If using unconjugated primary antibodies, 100 µl secondary antibody diluted in CAS-Block + 0.2% Triton X-100 was added to cells, and cells were incubated at 4℃ for 50 min. Subsequently, 3 ml 1X permeabilization buffer was added to the cells, and cells were spun at 1500 rpm for 5 min. The supernatant was discarded. Cells were resuspended in 250 µl FACS buffer and analyzed on the flow cytometer.

## RNA-seq and qPCR

RNA was isolated using the RNeasy Mini Kit (Qiagen, 74106) as per the manufacturer's instructions. To obtain N = 3 for the HPDE6c7 scr gRNA control, RNA was isolated on three different days of

subsequent passages. For qPCR, cDNA was prepared using the SuperScript III First Strand synthesis kit (Thermo Fisher Scientific, 18080085) using 500 ng of RNA and random hexamers. qPCR was performed using FastStart Universal SYBR Green mix (Sigma, 4913914001). RNA expression of target genes was normalized to GAPDH. qPCR primer sequences are included in *Supplementary file 4*.

For RNA-seq, a stranded mRNA library prep was prepared using PolyA capture and paired-end sequencing was performed by Novogene. 40 million reads were sequenced for each sample. Quality of raw FASTQ sequences was assessed using FASTQC. To process RNA-seq libraries, adaptor sequences were trimmed using Cutadapt version 1.14 (requiring a length >10 nucleotides after trimming) and quality-filtered by requiring all bases to have a minimum score of 20 (-m 20 -q 20). Only reads that passed the quality or length threshold on both strands were considered for mapping. Reads were aligned to the human genome GRCh38 (hg38) with the STAR Aligner (version 020201). Ensembl reference annotation version 89 was used to define gene models for mapping quantification. Uniquely mapped reads for each gene model were produced using STAR parameter '–quant-Mode GeneCounts.' Differential expression analysis was performed in R using DESeq2 (v.1.16.0) with the default parameters, including the Cook's distance treatment to remove outliers. The RNA-seq and scRNA-seq datasets were deposited to GEO (GEO accession #GSE159343).

## Cell culture assays

HPDE6c7 cells (RRID:CVCL_0P38) were authenticated by ATCC and tested negative for mycoplasma using a kit from InvivoGen (rep-pt1). HPDE6c7 cells (*Furukawa et al., 1996*) were cultured in DMEM (Life Technologies 11995073), 10% FBS (Corning, 35011CV), 1X Penicillin:Streptomycin solution (Corning, 30-002-CI). For cell counting, 25,000 cells were seeded in a sterile six-well TC-treated plate (Corning, 353046). Values depicted for all cell culture experiments represent the average of at least three independent experiments.

For carbonic anhydrase activity assays, cell lysates were prepared using standard protocols and cell lysis buffer (Cell Signaling Technologies, 9803S) containing 100 mM PMSF, 1X cOmplete Protease Inhibitor Cocktail (Roche, 11697498001), and 1X PhosSTOP (Sigma Aldrich, 4906845001). Carbonic anhydrase activity was measured using the carbonic anhydrase activity assay kit (Biovision, K472-100). For normalization, equal amounts of protein (10 µg) per sample were used in the assay. Protein concentration was determined using the Pierce BCA Protein Assay Kit (Thermo Fisher Scientific, 23225).

## Generation of stable knockout HPDE6c7 cell lines

For generation of stable knockouts, gRNAs were cloned into eSPCas-LentiCRISPR v2 (Genscript). gRNA sequences are included in *Supplementary file 3*. Each gRNA-containing plasmid was incorporated into lentivirus. HPDE6c7 cells were transduced with these lentiviruses, and cells expressing the gRNA-containing plasmid were selected for with puromycin. All cell culture experiments were performed using bulk transduced HPDE6c7 cells.

## Western blotting

Cell lysates were prepared using standard protocols and RIPA buffer (Thermo Fisher Scientific, 89901) containing 100 mM PMSF, 1X cOmplete Protease Inhibitor Cocktail (Roche, 11697498001), and 1X PhosSTOP (Sigma Aldrich, 4906845001). PVDF membranes were incubated with primary antibodies overnight at 4°C. After RT incubation with the appropriate HRP-conjugated secondary antibody for 1 hour, membranes were developed using SuperSignal West Pico PLUS Chemiluminescent Substrate (Thermo Scientific, 34580).

## Bioinformatics and statistical analysis

For cell culture studies, sample size was computed based on accepted scientific standards using a minimum of two CRISPR/Cas9-generated KO lines to control for off-target effects. For mouse experiments, sample size was computed based on the number of biological replicates required to obtain statistical significance. For *Gmnn* mouse model studies, mice with relevant genotypes were chosen randomly for PDL or control groups.

We used $p \leq 0.05$ as a cutoff for DEG inclusion for IPA and IPA upstream regulator analysis. Due to low cell number and high similarity, some comparisons did not yield an acceptable number of

statistically significant DEGs (≤25), and we used a relaxed p≤0.1 as a cutoff for these in order to identify more targets. GSEA was performed on the identified DEGs with the GSEA software (version 3.0) in the pre-ranked mode, with the Reactome pathway dataset (version 7.2). For analysis of published single-cell developmental biology datasets, GSM3140915 (E12.5 SW), GSM3140916 (E14.5 SW), GSM3140917 (E17.5 1 SW), and GSM3140918 (E17.5 2 SW) were used. The two E17.5 datasets were from the same animal and were merged. Ductal clusters were identified by expression of marker genes *Sox9*, *Krt19*, and *Epcam*.

Data are presented as mean ± SEM and were analyzed in GraphPad Prism or Microsoft Office Excel. Statistical significance was assumed at a p or q value of ≤ 0.05. p or q values were calculated with a t-test. For interpretation of statistical t-test results, *=p or q value ≤ 0.05, **=p or q value ≤ 0.01, ***=p or q value ≤ 0.001, and ****=p or q value ≤ 0.0001. For all statistical analyses, outliers were identified and excluded using the Grubbs' outlier test (alpha = 0.05) or ROUT (Q = 10%).

# Acknowledgements

The authors thank Christina S. Chung, Debbie Ngow, and Melissa Campbell for their indispensable help with mouse colony maintenance, genotyping, and technical assistance. The authors wish to acknowledge Luc Baeyens and Michael S. German for helpful intellectual discussions. Graphical illustrations were developed in part by Jimmy Chen using BioRender.com. The *Gmnn* floxed mouse strain was generated in Rudolf Jaenisch's laboratory. The authors thank Audrey Parent for flow cytometry protocols and guidance. This work was supported by a Richard G. Klein Fellowship in Pancreas Development, Regeneration or Cancer (to AMH). AMH was additionally supported by F32 CA221114 and a Hirshberg Foundation for Pancreatic Cancer Research Seed Grant. AAR and GKF were supported by the UCSF Bakar ImmunoX Initiative. EAC was supported by R01 CA222862. Work in MH's laboratory was supported by R01 CA172045 and a grant from the Parker Institute for Cancer Immunotherapy (PICI). MH owns stocks/stock options in Viacyte, Encellin, Thymmune, Endo-Crine, and Minutia.He also serves as SAB member to Thymmune and Encellin and is co-founder, SAB member, and board member for EndoCrine and Minutia.

# Additional information

## Funding

| Funder | Grant reference number | Author |
|---|---|---|
| National Cancer Institute | F32 CA221114 | Audrey M Hendley |
| Hirshberg Foundation for Pancreatic Cancer Research | Seed Grant | Audrey M Hendley |
| National Cancer Institute | R01 CA222862 | Eric A Collisson |
| National Cancer Institute | R01 CA172045 | Matthias Hebrok |
| Parker Institute for Cancer Immunotherapy | PICI Opportunity Grant | Matthias Hebrok |
| UCSF Foundation | UCSF Bakar ImmunoX Initiative | Arjun A Rao Gabriela K Fragiadakis |

The funders had no role in study design, data collection and interpretation, or the decision to submit the work for publication.

## Author contributions

Audrey M Hendley, Conceptualization, Data curation, Formal analysis, Supervision, Validation, Investigation, Methodology, Writing - original draft; Arjun A Rao, Sudipta Ashe, Jennifer A Smith, Simone Giacometti, Conceptualization, Data curation, Software, Formal analysis, Supervision, Validation, Investigation, Visualization, Methodology, Writing - review and editing; Laura Leonhardt, Conceptualization, Data curation, Formal analysis, Validation, Investigation, Methodology, Writing - review and editing; Xianlu L Peng, Conceptualization, Data curation, Software, Formal analysis, Validation,

Investigation, Visualization, Methodology; Honglin Jiang, Investigation, Methodology; David I Berrios, Lucia Y Li, Jonghyun Lee, Investigation; Mathias Pawlak, Conceptualization, Validation, Investigation, Methodology; Eric A Collisson, Matthias Hebrok, Resources, Supervision, Funding acquisition, Writing - review and editing; Mark S Anderson, Gabriela K Fragiadakis, Jen Jen Yeh, Valerie M Weaver, Resources, Supervision, Funding acquisition; Chun Jimmie Ye, Conceptualization, Resources, Supervision, Funding acquisition, Visualization, Writing - review and editing; Grace E Kim, Conceptualization, Data curation, Formal analysis, Supervision, Validation, Investigation, Visualization, Methodology

### Author ORCIDs

Audrey M Hendley ![ORCID] https://orcid.org/0000-0003-0289-2092
Eric A Collisson ![ORCID] http://orcid.org/0000-0001-8037-9388
Mark S Anderson ![ORCID] http://orcid.org/0000-0002-3093-4758
Matthias Hebrok ![ORCID] https://orcid.org/0000-0002-3833-8862

### Ethics

Animal experimentation: This study was performed in strict accordance with the recommendations in the Guide for the Care and Use of Laboratory Animals of the National Institutes of Health. All of the animals were handled according to approved Institutional Animal Care and Use Committee (IACUC) protocols (AN170192) of the University of California San Francisco.

### Decision letter and Author response

Decision letter https://doi.org/10.7554/eLife.67776.sa1
Author response https://doi.org/10.7554/eLife.67776.sa2

## Additional files

### Supplementary files

• Supplementary file 1. Expression scoring of markers of subpopulations of pancreatic duct cells. This table depicts a summary of expression scoring of selected markers for subpopulations of pancreatic duct cells in mouse and human tissue. Homogeneous refers to an observed uniform expression level and pattern within a particular ductal cell type. Heterogeneous means that either the observed expression level or pattern varies among cells within a particular ductal cell type.

• Supplementary file 2. The list of antibodies used in this study.

• Supplementary file 3. The list of gRNA sequences used in this study.

• Supplementary file 4. The list of qPCR primer sequences used in this study.

• Transparent reporting form

### Data availability

Sequencing data have been deposited in GEO under accession code GSE159343.

The following dataset was generated:

| Author(s) | Year | Dataset title | Dataset URL | Database and Identifier |
|---|---|---|---|---|
| Hendley AM, Rao AA, Leonhardt L, Ashe S, Smith JA, Giacometti S, Peng XL, Jiang H, Berrios DI, Pawlak M, Li LY, Lee J, Collisson EA, Anderson MS, Fragiadakis GK, Yeh JJ, Ye CJ, Kim GE, Weaver VM, Hebrok M | 2021 | Single cell RNA-seq defines novel heterogeneity within the pancreatic ductal tree | https://www.ncbi.nlm.nih.gov/geo/query/acc.cgi?acc=GSE159343 | NCBI Gene Expression Omnibus, GSE159343 |

The following previously published dataset was used:

| Author(s) | Year | Dataset title | Dataset URL | Database and Identifier |
|---|---|---|---|---|
| Byrnes LE, Wong DM, Subramaniam M, Meyer NP, Gilchrist CL, Knox SM, Tward AD, Ye CJ, Sneddon JB | 2018 | Lineage dynamics of pancreatic development at single cell resolution | https://www.ncbi.nlm.nih.gov/geo/query/acc.cgi?acc=GSE101099 | NCBI Gene Expression Omnibus, GSE101099 |

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
