## [Decision Letter]

**Acceptance summary:**

In this study, the authors present a high-resolution single-cell transcriptomic atlas of the pancreatic ductal tree. Their analysis unveils important heterogeneity within the pancreatic ductal populations and identifies unique cellular states. Overall, the results presented here suggest there are distinct functional roles for subpopulations of ductal cells in the maintenance of duct cell identity that could have broader implications for chronic pancreatic inflammation. Furthermore, the detailed analysis of the pancreatic subpopulations could also be relevant in the context of cancer biology and provides novel information that may help in elucidating the transition from pancreatitis to pancreatic cancer and/or different predispositions to cancer.

**Decision letter after peer review:**

Thank you for submitting your article "Single cell transcriptome analysis defines heterogeneity of the murine pancreatic ductal tree" for consideration by *eLife*. Your article has been reviewed by 3 peer reviewers, one of whom is a member of our Board of Reviewing Editors, and the evaluation has been overseen by Didier Stainier as the Senior Editor. The reviewers have opted to remain anonymous.

Essential revisions:

1. The extensive text describing the gene expression analyses and the different clusters should be streamlined to emphasize the take home message. This could be accompanied by a schematic figure that includes representative genes in each cluster. There should also be more limited "conclusions" made from the gene expression data; many of these are overstated.

2. Provide additional data to support the presence of an "EMT axis" or downplay this conclusion.

3. Provide rationale for choosing Anxa3, SPP1 and Geminin for further analysis and indicate why analysis Anxa3 wasn't pursued further. Also several other genes were mentioned in this section with reasoning as to why they weren't pursued further, but it isn't clear why they were chosen in the first place.

4. As indicated by reviewer 2, there are several parts of the text that should include rationale for the experiments. For instance, choice of cell line, etc.

5. Although a lot of effort went into generating the mutant models for functional analyses of SPP1 and Geminin, the analysis is incomplete. In their current form, the phenotypes of the different mutants show potential, but the analysis is still incomplete. The authors should acknowledge this and indicate that future experiments will be necessary to fully understand their roles.

6. The proliferation analysis of the KO HPDE lines is not convincing and the authors should provide cell counting over a time course.

7. Improve the IF staining as indicated by reviewer 3.

*Reviewer #1 (Recommendations for the authors):*

1. The extensive text describing the gene expression analyses and different clusters should be streamlined to emphasize the take home message. This could be accompanied by a schematic figure that includes representative genes in each cluster.

2. Provide additional data to support the presence of an "EMT axis" or downplay this conclusion.

3. Provide rationale for choosing Anxa3, SPP1 and Geminin for further analysis and indicate why analysis Anxa3 wasn't pursued further. Also several other genes were mentioned in this section with reasoning as to why they weren't pursued further, but it isn't clear why they were chosen in the first place.

4. Although a lot of effort went into generating the mutant models for functional analyses of SPP1 and Geminin, the analysis is incomplete seem to be added as an afterthought at the end of the study. In their current form, the phenotypes of the different mutants show potential, but the analysis is too incomplete and actually compromise the first half of the paper. The authors should either flesh out one of the studies or remove them from the paper.

5. The proliferation analysis of the KO HPDE lines is not convincing – the authors would need to provide cell counting over a time course.

6. Additional quantification for all of the experiments is required. For SPP, the number of Ngn3/*Sox9* cells and endocrine cells formed vs exocrine. For Geminin, the percent of cells showing H2AX loci should be quantified.

*Reviewer #2 (Recommendations for the authors):*

1. The expression profile of Kcn3 and Sparc does not appear to correlate well with cluster 2 (figure 2C).

2. IPA analyses are presented selectively eg for cluster 2 arginine metabolism pathways are mentioned but not cysteine metabolism which shows stronger p-value. In any case, these are all speculations and as such they are not appropriate for main figures (figure 2D).

3. The authors suggest that cluster 5 of cycling cells represents cells of clusters 3 and 4 but the distance in the UMAP from cluster 4 and the poor Pearson correlation (figure 3B) do not support the relationship with cluster 4. Also the pseudo-temporal ordering suggested that clusters 3 and 5 are not related to the other clusters.

4. The authors write in the beginning of the 2nd paragraph in page 9 : 'Our analysis thus far has identified and validated multiple transcriptional programs.… ´. This is a clear over interpretation – bioinformatics analysis alone cannot validate transcriptional programs.

5. Functional analyses of some marker genes was undertaken using immortalized HPDE cells (page 9). Apart from the general concern for the validity of results using immortalized cell lines, it is not explained what is the rationale for selecting this particular cell line and why Anxa3 was selected. It is also not clear why they mention here that PAH and WFDC3 are not expressed in this cell line. If these genes are important should they not be analyzed in another context ?

6. It is mentioned that SPP1 knockout lines had prominent filopodia but this is not shown or commented upon – what does it mean ? There is evidence from the RNA Seq for EMT and it will be essential if the authors further establish that by immunofluorescence analyses.

7. The evidence for increased proliferation of SPP1 KO lines is not strong enough. The CGT assays detect endogenous ATP levels and such increase does not necessarily mean increased cell proliferation. Similarly, increased relative spheroid size does not mean much if it is accompanied by a reduction in the number of spheroids. Cell counts at successive passages would be necessary.

8. Even if there is increase in the proliferation rate of this transformed line as well as taking into account the RNA seq data, the conclusion that loss of SPP1 leads to a progenitor like state is not warranted. Nothing is mentioned on the expression at the transcript or protein level of established markers such as eg NKX6.1, PDX1, HNF3B etc. These are straightforward experiments to bolster the analysis.

9. The authors proceed with further analysis of two of the four knock out SPP1 lines. The criteria for selecting these two are not clear.

10. The authors attempt to document SPP1 downregulation, changes in Vimentin and CK19 expression by immunofluorescence (figure 6J, K). Controls (scr gRNA) are shown in different magnification than knock down lines. This makes it harder to document changes and it is questionable practice. I am not convinced by the claim for changes in SPP1 and Vimentin expression for either of the two lines. Regarding CK19, only one of the two stainings appears convincing. This is problematic particularly regarding SPP1 – this is supposed to be a nearly complete KO (figure 6B). It may be a matter of levels in which case IF is not appropriate. Flow cytometry profiles could resolve this.

11. The differentiation potential of SPP1 knocked out cells was then tested five days after subcutaneous transplantation by immunofluorescence. The Glucagon and Ngn3 stainings are not convincing. Also, whereas there appear to be several C-pep+ cells the NkX6.1+ and PDX1+ among them are very scarce. This would not be expected from a differentiation into genuine b cells.

12. Geminin is expressed in very few ductal cells even after PDL (0.05%, figure 8B). It is not explained what are the quantitated ´γ-H2AX+ foci´. Do these correspond to γ-H2AX +cells ? It should be stated clearly; what is the % of cells that are γ-H2AX+ ?

*Reviewer #3 (Recommendations for the authors):*

Few concerns that should be addressed are listed below:

– Some of immunofluorescence (IF) staining experiments could be optimized and replaced with better image qualities. For instance, in Figure 7E Pdx1+ cells are not visible in the transplanted HPDE cells stained for Synaptophysin/Pdx1/C-peptide. Similarly, in the IF on human pancreatic tissue (see Figure 8A) the distribution of Geminin is not clearly visible. The IF should be repeated and the images showing the individual channels should be included?

– The study reports some interesting observations on the pancreatobiliary and intrapancreatic bile duct cells. It would be helpful to include spatial characterization of these cells or a schematics to indicate the location of these populations. Are they only in the intrapancreatic bile common duct?

---

## [Author Response]

Essential revisions:1. The extensive text describing the gene expression analyses and the different clusters should be streamlined to emphasize the take home message. This could be accompanied by a schematic figure that includes representative genes in each cluster. There should also be more limited "conclusions" made from the gene expression data; many of these are overstated.

We agree with these comments and have reduced the description of the duct clusters as much as possible. We have retained the points we believe are important including those which are referenced in the Discussion section as well as remarks that necessarily reference the figure panels, datasets, and tables in chronological order. We retained a longer description of cluster 4 as we believe the points are important to annotate this population as pancreatobiliary cells. We have also provided additional data to support several bioinformatics analyses as detailed below. Additionally, we tried to modify the language regarding the bioinformatics analyses that is not supported with experiments to reflect speculation rather than conclusions. As requested, we have included a new schematic word cloud of the top differentially expressed genes in each duct cluster.

2. Provide additional data to support the presence of an "EMT axis" or downplay this conclusion.

We agree that the data supporting the EMT axis, which was limited to the CK19 and Vimentin co-IF in Figure 5F, could be strengthened. We examined additional EMT markers in human and mouse ductal cells and found heterogenous expression of SNAI1 and fibronectin in both mouse and human ductal cells. Fibronectin was scarce while SNAI1 expression was abundant in both mouse and human duct cells. Due to lack of space in the figures in this part of the manuscript to add additional data, we included images for only the human duct cells for SNAI1 and fibronectin co-IF in Figure 5—figure supplement 1E.

With regards to *SPP1*, a gatekeeper of the EMT axis, we have provided additional evidence for EMT in the HPDE6c7 *SPP1* KO lines. We performed qPCR for epithelial and mesenchymal genes in HPDE6c7 *SPP1* KO lines and the HPDE6c7 scr gRNA control and included this data in Figure 6—figure supplement 1A-B. IPA analysis data already included in Figure 6-source data 2 shows the Regulation of the Epithelial-Mesenchymal Transition Pathway is significantly enriched from the RNA-seq data of HPDE6c7 *SPP1* KO lines vs. HPDE6c7 scr gRNA control. Additionally, we have added flow cytometry data of epithelial and mesenchymal markers for HPDE6c7 *SPP1* KO lines and the HPDE6c7 scr gRNA control in Figure 6—figure supplement 1C.

3. Provide rationale for choosing Anxa3, SPP1 and Geminin for further analysis and indicate why analysis Anxa3 wasn't pursued further. Also several other genes were mentioned in this section with reasoning as to why they weren't pursued further, but it isn't clear why they were chosen in the first place.

We have added rationale for choosing *Anxa3*, *Spp1*, and *Gmnn* for further analysis in the section titled, “*SPP1* is required for mature human pancreas duct cell identity.” *Anxa3* was not pursued further because we wanted to focus our functional studies on the strongest phenotypes observed. Rationale for choosing Geminin to pursue in *in vivo* pancreatitis studies is provided under the section titled “Geminin safeguards against accumulation of DNA damage in mouse ductal cells in the setting of chronic pancreatitis” – “One marker of the workhorse population of pancreatic duct cells *Gmnn* has previously been associated with chronic inflammatory diseases such as asthma (59). We therefore queried its role in pancreas inflammatory disease.” We mentioned that *PAH* and *WFDC3* were not expressed in HPDE6c7 cells as reasoning for not pursuing them further in functional studies because we examined these markers of duct subpopulations by IHC in Figure 2—figure supplement 1-2. We agree with the reviewer that it isn’t necessary to mention this because we don’t pursue these genes further and have removed this sentence.

4. As indicated by reviewer 2, there are several parts of the text that should include rationale for the experiments. For instance, choice of cell line, etc.

We have added rationale for choice of the HPDE6c7 cell line in the section entitled “*SPP1* is required for mature human pancreas duct cell identity.”

5. Although a lot of effort went into generating the mutant models for functional analyses of SPP1 and Geminin, the analysis is incomplete. In their current form, the phenotypes of the different mutants show potential, but the analysis is still incomplete. The authors should acknowledge this and indicate that future experiments will be necessary to fully understand their roles.

We have acknowledged that our analysis of the *SPP1* and *Geminin* mutants is incomplete and suggest that further experiments are required to fully understand their roles in the Discussion section.

6. The proliferation analysis of the KO HPDE lines is not convincing and the authors should provide cell counting over a time course.

Please see comment #7 from Reviewer #2.

7. Improve the IF staining as indicated by reviewer 3.

Please see comment #1 from Reviewer #3.

Reviewer #1 (Recommendations for the authors):1. The extensive text describing the gene expression analyses and different clusters should be streamlined to emphasize the take home message. This could be accompanied by a schematic figure that includes representative genes in each cluster.

Please see comment #1 from Essential revisions.

2. Provide additional data to support the presence of an "EMT axis" or downplay this conclusion.

Please see comment #2 from Essential revisions.

3. Provide rationale for choosing Anxa3, SPP1 and Geminin for further analysis and indicate why analysis Anxa3 wasn't pursued further. Also several other genes were mentioned in this section with reasoning as to why they weren't pursued further, but it isn't clear why they were chosen in the first place.

Please see comment #3 from Essential revisions.

4. Although a lot of effort went into generating the mutant models for functional analyses of SPP1 and Geminin, the analysis is incomplete seem to be added as an afterthought at the end of the study. In their current form, the phenotypes of the different mutants show potential, but the analysis is too incomplete and actually compromise the first half of the paper. The authors should either flesh out one of the studies or remove them from the paper.

We followed the guidance from the editor and indicated in the Discussion section that further analyses are required to understand the roles of *SPP1* and Geminin. The authors are open to removing the *SPP1* study using NSG mice, the pancreatitis study of Geminin, or both from the paper if the editor and reviewers feel that the remaining data collectively constitute an improved article suitable for publication in *eLife*.

5. The proliferation analysis of the KO HPDE lines is not convincing – the authors would need to provide cell counting over a time course.

Please see comment #7 from Reviewer #2.

6. Additional quantification for all of the experiments is required. For SPP, the number of Ngn3/Sox9 cells and endocrine cells formed vs exocrine. For Geminin, the percent of cells showing H2AX loci should be quantified.

We understand and share the reviewer’s caution regarding the formation of cells with endocrine properties. We cautiously mention that a small subset of cells have an endocrine-like appearance demonstrating expression of NKX6.1, synaptophysin, PDX1, and c-peptide (Figure 7), while also noting that being able to demonstrate full transdifferentiation to mature endocrine cells would require functional and full transcriptome analyses that, due to the low number of such cells, are beyond the scope of this manuscript. We have included a quantification of the c-peptide/NKX6.1/PDX1 positive cells in Figure 7F that supports the notion of cells with endocrine potential forming while also indicating that these are rare events. The number of Ngn3/*Sox9* cells is also quite rare as indicated in the IF pictures in Figure 7. Considering that we provide quantification of the more relevant c-peptide/NKX6.1/PDX1 positive cells, we just mention the equally rare nature of the NGN3/SOX9 cells without quantification. Additionally, with regards to quantification of γ-H2AX foci in Geminin mutant samples, please see comment #12 from Reviewer #2.

Reviewer #2 (Recommendations for the authors):1. The expression profile of Kcn3 and Sparc does not appear to correlate well with cluster 2 (figure 2C).

We apologize for not describing the nature of Cluster 2 more clearly. Cluster 2 is characterized by low level or lack of expression of multiple ductal cell markers (*Cftr*, *Kcne3*, *Sparc*, *Mmd2, Krt7*) found in other clusters (Figure 2B-C and Figure 1—figure supplement 1G). Most markers in Cluster 2 are also present in Cluster 0 (please see Figure 3D). When comparing Clusters 0 vs 2, only 9 DEGs come up. None of these genes have an expression pattern specific for Cluster 2. Characterizing a cluster by lack of specific gene expression is something with precedent in the single cell sequencing field.

2. IPA analyses are presented selectively eg for cluster 2 arginine metabolism pathways are mentioned but not cysteine metabolism which shows stronger p-value. In any case, these are all speculations and as such they are not appropriate for main figures (figure 2D).

We acknowledge that the pathway analysis is not conclusive, but would argue that it provides important information regarding the diverse nature of the various clusters. Figure 2D shows the top 8 deregulated pathways when comparing a cluster to all other clusters using IPA. We have removed the mention of arginine metabolism for cluster 2 in the text, in an effort to streamline the extensive text describing the gene expression analyses and the different clusters as suggested by the editor and Reviewer #1.

3. The authors suggest that cluster 5 of cycling cells represents cells of clusters 3 and 4 but the distance in the UMAP from cluster 4 and the poor Pearson correlation (figure 3B) do not support the relationship with cluster 4. Also the pseudo-temporal ordering suggested that clusters 3 and 5 are not related to the other clusters.

We agree with the comment and have removed this suggestion in an effort to streamline the extensive text describing the gene expression analyses and the different clusters as also suggested by the editor and Reviewer #1.

4. The authors write in the beginning of the 2nd paragraph in page 9 : 'Our analysis thus far has identified and validated multiple transcriptional programs.… ´. This is a clear over interpretation – bioinformatics analysis alone cannot validate transcriptional programs.

By writing “our analysis thus far”, we are referring to both the bioinformatics analysis that identified the transcriptional programs as well as our IHC/IF validation of protein expression of several markers of these murine ductal subpopulations identified in our sc-RNA-seq dataset depicted in Figure 2—figure supplement 1-3. We have reworded this sentence.

5. Functional analyses of some marker genes was undertaken using immortalized HPDE cells (page 9). Apart from the general concern for the validity of results using immortalized cell lines, it is not explained what is the rationale for selecting this particular cell line and why Anxa3 was selected. It is also not clear why they mention here that PAH and WFDC3 are not expressed in this cell line. If these genes are important should they not be analyzed in another context ?

Please see comment #3 from Essential revisions.

6. It is mentioned that SPP1 knockout lines had prominent filopodia but this is not shown or commented upon – what does it mean ? There is evidence from the RNA Seq for EMT and it will be essential if the authors further establish that by immunofluorescence analyses.

The filipodia are evident in the higher magnification images in Figure 6D. We have added yellow arrows to make them clearly visible to the reader. In Figure 6E, we show increased proliferation in HPDE6c7 *SPP1* KO cells when compared to HPDE6c7 scr gRNA cells. We mention filipodia in the manuscript because they are often prominent in migratory cells, serving as sensors of the environment and playing roles in cell migration and wound healing. The prominent filipodia in HPDE6c7 *SPP1* KO cells hint at potentially enhanced migration, in addition to proliferation, and we plan to pursue follow up studies to better characterize the migratory features of these cell lines, which are out of the scope of this manuscript. Additionally, please see comment #2 from Essential revisions.

7. The evidence for increased proliferation of SPP1 KO lines is not strong enough. The CGT assays detect endogenous ATP levels and such increase does not necessarily mean increased cell proliferation. Similarly, increased relative spheroid size does not mean much if it is accompanied by a reduction in the number of spheroids. Cell counts at successive passages would be necessary.

We agree with the reviewer’s comment and have removed the CTG assays and spheroid data from the manuscript and replaced it with the requested cell counting experiment in Figure 6E. Additionally, many pathways involved in positive regulation of cell cycle signaling were identified in the GSEA analysis for HPDE6c7 *SPP1* KO cell lines (Figure 6—figure supplement 1D).

8. Even if there is increase in the proliferation rate of this transformed line as well as taking into account the RNA seq data, the conclusion that loss of SPP1 leads to a progenitor like state is not warranted. Nothing is mentioned on the expression at the transcript or protein level of established markers such as eg NKX6.1, PDX1, HNF3B etc. These are straightforward experiments to bolster the analysis.

The reviewer brings up another important point. To strengthen our assessment of a progenitor-like state in HPDE6c7 *SPP1* KO cell lines, we have performed qPCR for pancreatic progenitor markers *Pdx1*, *Ptf1a*, *Gata4*, *Gata6*, *Prox1*, *Myt1*, and *Rxrg* and added our analysis to Figure 6K. Towards support for a progenitor-like state for HPDE6c7 *SPP1* KO cell lines, we also reference the GSEA analysis in Figure 6—figure supplement 1E-F.

9. The authors proceed with further analysis of two of the four knock out SPP1 lines. The criteria for selecting these two are not clear.

To avoid bias, the selection of HPDE6c7 *SPP1* KO lines for experiments was random. All HPDE6c7 *SPP1* KO lines display similar levels of *SPP1* knockdown (Figure 6B) and similar levels of gene expression for all genes, according to the normalized counts file for the RNA-seq data. When 2 or 3 lines are selected randomly for experiments, they show similar results; thus, we believe this selection criteria is unbiased, and a valid approach.

10. The authors attempt to document SPP1 downregulation, changes in Vimentin and CK19 expression by immunofluorescence (figure 6J, K). Controls (scr gRNA) are shown in different magnification than knock down lines. This makes it harder to document changes and it is questionable practice. I am not convinced by the claim for changes in SPP1 and Vimentin expression for either of the two lines. Regarding CK19, only one of the two stainings appears convincing. This is problematic particularly regarding SPP1 – this is supposed to be a nearly complete KO (figure 6B). It may be a matter of levels in which case IF is not appropriate. Flow cytometry profiles could resolve this.

It is important to point out that controls and KO’s are shown in the same magnification. The HPDE6c7 *SPP1* KO cell lines are smaller cells with a different morphology, as shown in Figure 6D. Knockdown of *SPP1* in HPDE6c7 *SPP1* KO cell lines is shown by WB in Figure 6B and normalized RNA-seq counts data in Figure 6G. We have replaced the Vimentin and OPN immunocytochemistry images in Figure 6I for the HPDE6c7 *SPP1* KO cell lines with better quality images. We believe the issue with the previous images was autofluorescence. We have added flow cytometry profiles to Figure 6—figure supplement 1C, and removed one of the GSEA plots to make space for them.

11. The differentiation potential of SPP1 knocked out cells was then tested five days after subcutaneous transplantation by immunofluorescence. The Glucagon and Ngn3 stainings are not convincing. Also, whereas there appear to be several C-pep+ cells the NkX6.1+ and PDX1+ among them are very scarce. This would not be expected from a differentiation into genuine b cells.

Please see comment #6 from Reviewer #1.

12. Geminin is expressed in very few ductal cells even after PDL (0.05%, figure 8B). It is not explained what are the quantitated ´γ-H2AX+ foci´. Do these correspond to γ-H2AX +cells ? It should be stated clearly; what is the % of cells that are γ-H2AX+ ?

In the methods section under histology and immunostaining, we write, “For quantification of BrdU, cleaved caspase 3, Geminin, Ki67, γ*-*H2AX, and ATR, at least 60 cells from 3 different ducts were analyzed.” These 60 cells were not chosen based on any specific criteria. Quantification of γ*-*H2AX foci include duct cells with zero foci. We have added this to the methods section.

Reviewer #3 (Recommendations for the authors):Few concerns that should be addressed are listed below:– Some of immunofluorescence (IF) staining experiments could be optimized and replaced with better image qualities. For instance, in Figure 7E Pdx1+ cells are not visible in the transplanted HPDE cells stained for Synaptophysin/Pdx1/C-peptide. Similarly, in the IF on human pancreatic tissue (see Figure 8A) the distribution of Geminin is not clearly visible. The IF should be repeated and the images showing the individual channels should be included?

We have repeated the IF staining experiments for Figure 7E and Figure 8A and replaced most images. For Figure 7E Synaptophysin/Pdx1/C-peptide immunostaining, after repeating the IF staining, the images already in the figure for HPDE6c7 *SPP1* gRNA6 remained the best quality images we were able to generate, and thus, we left these images in the panel. We have replaced the images for HPDE6c7 *SPP1* gRNA8 to show an area with a higher cell density. To clarify the location of PDX1+, synaptophysin+, and c-peptide+ triple positive cells, we have included yellow arrows in the revised panel. HPDE6c7 scr gRNA cells are uniformly negative for PDX1, synaptophysin, and c-peptide.

– The study reports some interesting observations on the pancreatobiliary and intrapancreatic bile duct cells. It would be helpful to include spatial characterization of these cells or a schematics to indicate the location of these populations. Are they only in the intrapancreatic bile common duct?

We agree that spatial information regarding markers of the pancreatobiliary and intrapancreatic bile ducts is an important question. Our examination of markers characterizing these populations transcriptionally: Acetylated α tubulin (*Tuba1a*) for Cluster 3 (intrapancreatic bile duct cells) and CXCL5 for Cluster 4 (pancreatobiliary cells) showed positive cells in both pancreatobiliary and intrapancreatic bile ducts (Figure 2—figure supplement 2-3) in mouse and human. While *Tuba1a* is clearly enriched in cluster 3 (Figure 2-source data 1), the feature plot in Figure 2—figure supplement 2D depicts expression of *Tuba1a* also in cluster 4.

The markers we selected for evaluation of clusters 3 and 4 didn’t suggest a clear spatial distinguishment from intrapancreatic bile duct and pancreatobiliary cells. We cannot be certain of the spatial location of every cell in clusters 3 or 4 and thus, a schematic showing spatial location of these populations would require additional comprehensive marker assessment.